



# Site specific parameterizations of longwave radiation

Giuseppe Formetta[1], Marialaura Bancheri[2], Olaf David [3] and Riccardo Rigon [2]

[1]Centre for Ecology & Hydrology, Crowmarsh Gifford, Wallingford, UK

[2]Dipartimento di Ingegneria Civile Ambientale e Meccanica, Universita' degli Studi di Trento,

Italy

[3]Dept. of Civil and Environmental Engineering, Colorado State University, Fort Collins, CO,

USA

May 30, 2016

## Abstract

In this work ten algorithms for estimating downwelling longwave atmospheric radiation ($L_\downarrow$) and one for upwelling longwave radiation ($L_\uparrow$) are integrated into the hydrological model JGrass-NewAge. The algorithms are tested against energy flux measurements available for twenty-four sites in North America to assess their reliability. These new JGrass-NewAge model components are used i) to evaluate the performances of simplified models (SMs) of $L_\downarrow$, as presented in literature formulations, and ii) to determine by automatic calibration the site-specific parameter sets for SMs of $L_\downarrow$. For locations where calibration is not possible because of a lack of measured data, we perform a multiple regression using on-site variables, such as mean annual air temperature, relative humidity, precipitation, and altitude. The regressions are verified through a leave-one-out cross validation, which also gathers information about the possible errors of estimation. Most of the SMs, when executed with parameters derived from the multiple regressions, give enhanced performances compared to the corresponding literature formulation. A sensitivity analysis is carried out for each SM to understand how small variations of a given parameter influence SM performance. Regarding the $L_\downarrow$ simulations, the Brunt (1932) and Idso (1981) SMs, in their literature formulations, provide the best performances in many of the sites. The site-specific parameter calibration improves SM performances compared to their literature formulations. Specifically, the root mean square error (RMSE) is almost halved and the Kling Gupta efficiency is improved at all sites.

The $L_\uparrow$ SM is tested by using three different temperatures (surface soil temperature, air temperature at 2 m elevation, and soil temperature at 4 cm depth) and model performances are then assessed. Results show that the best performances are achieved using the surface soil temperature and the air temperature.

Models and regression parameters are available for any use, as specified in the paper.





## 1 Introduction

Longwave radiation (1-100 $\mu$m) is an important component of the radiation balance on earth and it affects many phenomena, such as evapotranspiration, snow melt (Plüss and Ohmura, 1997), glacier evolution (MacDonell et al., 2013), vegetation dynamics (Rotenberg et al., 1998), plant respiration, and primary productivity (Leigh Jr, 1999). Longwave radiation is usually measured with very expensive pyrgeometers, but these are not normally available in basic meteorological stations, even though an increasing number of projects has been developed to fill the gap, Augustine et al. (2000), as seen in Augustine et al. (2005) and Baldocchi et al. (2001). The use of satellite products to estimate longwave solar radiation is increasing (GEWEX, Global Energy and Water cycle Experiment, ISCCP the International Satellite Cloud Climatology Project) but they have too coarse a spatial resolution for many hydrological uses. Therefore, models have been developed to solve energy transfer equations and compute radiation at the surface, e.g. Key and Schweiger (1998), Kneizys et al. (1988). These physically based and fully distributed models provide accurate estimates of the radiation components. However, they require input data and model parameters that are not easily available. To overcome this issue, simplified models (SM), which are based on empirical or physical conceptualizations, have been developed to relate longwave radiation to atmospheric proxy data such as air temperature, deficit of vapor pressure, and shortwave radiation. They are widely used and provide clear sky (e.g. Ångström (1915); Brunt (1932); Idso and Jackson (1969)) and all-sky estimations of downwelling, $L_\downarrow$, and upwelling, $L_\uparrow$, longwave radioation(e.g. Brutsaert (1975); Iziomon et al. (2003a)).

SM performances have been assessed in many studies by comparing measured and modeled $L_\downarrow$ at hourly and daily time-steps (e.g. Sugita and Brutsaert (1993b); Iziomon et al. (2003b); Juszak and Pellicciotti (2013)). Hatfield et al. (1983) was among the first to present a comparison of the most used SMs in an evaluation of their accuracy. It tested seven clear-sky algorithms using atmospheric data from different stations in the United States. So In order to validate the SMs under different climatic conditions, they performed linear regression analyses on the relationship between simulated and measured $L_\downarrow$ for each algorithm. The results of the study show that the best models were Brunt (1932), Brutsaert (1975) and Idso (1981). Flerchinger et al. (2009) made a similar comparison using more formulations (13) and a wider data-set from North America and China, considering all possible sky conditions. Finally, Carmona et al. (2014) evaluated the performance of six SMs, with both literature and site-specific formulations, under clear-sky conditions for the sub-humid Pampean region of Argentina. However, none of the above studies have provided a comprehensive set of open-source tools that are well documented and ready for practical use by other researchers and practitioners.

This paper introduces the LongWave Radiation Balance package (LWRB) of the JGrass-NewAGE modelling system Formetta et al. (2014a). LWRB implements 10 formulations for $L_\downarrow$ and one for $L_\uparrow$ longwave radiation. The package was systematically tested against measured $L_\downarrow$ and $L_\uparrow$ longwave radiation data from 24 stations across the USA, chosen from the 65 stations of the AmeriFlux Network. Unlike all previous works, the LWRB component follows the specifications of the Object Modeling System (OMS) framework, David et al. (2013). Therefore, it can use all of the JGrass-NewAge tools for the automatic calibration algorithms, data management





and GIS visualization, and it can be seamlessly integrated into various modeling solutions for the estimation of
water budget fluxes (Formetta et al., 2014a).
The paper is organized into five sections, with Section 1 being this introduction. Section 2 describes methodology, calibration and verification for the $L_\downarrow$ and $L_\uparrow$ models. Section 3 presents the study sites and the datasets
used. Section 4 presents the simulation results for $L_\downarrow$ and $L_\uparrow$ longwave radiation. It includes model verification
used. Section 4 presents the simulation results for $L_\downarrow$ and $L_\uparrow$ longwave radiation. It includes model verification
and calibration, sensitivity analysis and multiple regressions of the parameters against some explaining variables
for $L_\downarrow$. It also presents a verification of the $L_\uparrow$ model, which includes an assessment of the model performances
in predicting correct upwelling longwave $L_\uparrow$ radiation in using different temperatures (soil surface temperature,
air temperature, and soil temperature at 4 cm below surface). In Section 5 we present our conclusions.

## 2   Methodology

The SMs for $L_\uparrow$ [Wm$^{-2}$] and $L_\downarrow$ [Wm$^{-2}$] longwave radiation are based on the Stefan-Boltzmann equation:

$$L_\downarrow = \epsilon_{all-sky} \cdot \sigma \cdot T_a^4 \tag{1}$$

$$L_\uparrow = \epsilon_s \cdot \sigma \cdot T_s^4 \tag{2}$$

where $\sigma = 5.670 \cdot 10^{-8}$ [Kg s$^{-3}$ K$^{-4}$] is the Stefan-Boltzmann constant, $T_a$ [K] is the near-surface air
temperature, $\epsilon_{all-sky}$ [-] is the effective atmospheric emissivity, $\epsilon_s$ [-] is the soil emissivity and $T_s$ [K] is the
surface soil temperature. To account for the increase of $L_\downarrow$ in cloudy conditions, $\epsilon_{all-sky}$ [-] is formulated
according to eq. (3):

$$\epsilon_{all-sky} = \epsilon_{clear} \cdot (1 + a \cdot c^b) \tag{3}$$

where $c$ [-] is the clearness index and $a$ [-] and $b$ [-] are two calibration coefficients. Site specific values of $a$
and $b$ are presented in Brutsaert (1975), ($a$=0.22 and $b$=1), Iziomon et al. (2003a) ($a$ ranges between 0.25 and
0.4 and $b$=2) and Keding (1989) ($a$=0.183 and $b$=2.18). In our modeling system $a$ and $b$ are calibrated to fit
measurement data under all-sky conditions. The cloud cover fraction, $c$, can be estimated from solar radiation
measurements (Crawford and Duchon, 1999), from visual observations (Alados-Arboledas et al., 1995, Niemelä
et al., 2001), and from satellite data (Sugita and Brutsaert, 1993a) or it can be modeled as well. In this study
we use the formulation presented in Campbell (1985) and Flerchinger (2000), where $c$ is related to the clearness
index, (i.e. the ratio between the measured incoming solar radiation, $I_m$ [Wm$^{-2}$], and the theoretical solar
radiation computed at the top of the atmosphere, $I_{top}$ [Wm$^{-2}$]). This type of formulation needs a shortwave
radiation balance model to estimate $I_{top}$ and meteorological stations to measure $I_m$; also, it cannot estimate $c$
at night. In our application, the fact that the SMs are fully integrated into the JGrass-NewAge system allows
us to use the shortwave radiation balance model (Formetta et al., 2013 ) to compute $I_{top}$. Night-time values of
$c$ are computed with a linear interpolation between its values at the last hour of daylight and the first hour of



daylight on consecutive days. Ten SMs from literature have been implemented for the computation of $\epsilon_{clear}$.
Table 1 specifies assigned component number, component name, defining equation, and reference to the paper
from which it is derived. X, Y and Z are the parameters provided in literature for each model, listed in table 2.

| ID | Name | Formulation | Reference |
|---|---|---|---|
| 1 | Angstrom | $\epsilon_{clear} = X - Y \cdot 10^{Ze}$ | Angstrom [1918] |
| 2 | Brunt's | $\epsilon_{clear} = X + Y \cdot e^{0.5}$ | Brunt's [1932] |
| 3 | Swinbank | $\epsilon_{clear} = X \cdot 10^{-13} \cdot T_a^6$ | Swinbank [1963] |
| 4 | Idso and Jackson | $\epsilon_{clear} = 1 - X \cdot exp(-Y \cdot 10^{-4} \cdot (273 - T_a)^2)$ | Idso and Jackson [1969] |
| 5 | Brutsaert | $\epsilon_{clear} = X \cdot (e/T_a)^{1/Z}$ | Brutsaert [1975] |
| 6 | Idso | $\epsilon_{clear} = X + Y \cdot 10^{-4} \cdot e \cdot exp(1500/T_a)$ | Idso [1981] |
| 7 | Monteith and Unsworth | $\epsilon_{clear} = X + Y \cdot \sigma \cdot T_a^4$ | Monteith and Unsworth [1990] |
| 8 | Konzelmann | $\epsilon_{clear} = X + Y \cdot (e/T_a)^{1/8}$ | Konzelmann et al [1994] |
| 9 | Prata | $\epsilon_{clear} = [1 - (X + w) \cdot exp(-(Y + Z \cdot w)^{1/2})]$ | Prata [1996] |
| 10 | Dilley and O'Brien | $\epsilon_{clear} = X + Y \cdot (T_a/273.16)^6 + Z \cdot (w/25)^{1/2}$ | Dilley and O'Brien [1998] |

**Table 1:** Clear sky emissivity formulations: $T_a$ is the air temperatue [K], w $[kg/m^2]$ is precipitable water $= 4650 \, [e_0/T_a]$ and e [kPa] is screen-level water-vapour pressure.

The models presented in table 1 were proposed with coefficient values (X, Y, Z) strictly related to the location
in which the authors applied the model and where measurements of $L_\downarrow$ radiation were collected. Coefficients

reflect climatic, atmospheric and hydrological conditions of the sites, and are reported in Table 2.

| ID | Name | X | Y | Z |
|---|---|---|---|---|
| 1 | Angstrom | 0.83 | 0.18 | −0.07 |
| 2 | Brunt | 0.52 | 0.21 | [−] |
| 3 | Swinbank | 5.31 | [−] | [−] |
| 4 | Idso and Jackson | 0.26 | −7.77 | [−] |
| 5 | Brutsaert | 1.72 | 7 | [−] |
| 6 | Idso | 0.70 | 5.95 | [−] |
| 7 | Monteith and Unsworth | −119.00 | 1.06 | [−] |
| 8 | Konzelmann et al | 0.23 | 0.48 | [−] |
| 9 | Prata | 1.00 | 1.20 | 3.00 |
| 10 | Dilley and O'brien | 59.38 | 113.70 | 96.96 |

**Table 2:** Model parameter values as presented in their literature formulation.


The formulation of the $L_\uparrow$ requires the soil emissivity, which usually is a property of the nature of a surface,
and the surface soil temperature. Table 3 shows the literature values(Brutsaert, 2005) of the soil emissivity for
different surface types: $\epsilon_s$ varies from a minimum of 0.95 for bare soils to a maximum of 0.99 for fresh snow.

| Nature of surface | Emissivity |
|---|---|
| Bare soil (mineral) | $0.95 - 0.97$ |
| Bare soil (organic) | $0.97 - 0.98$ |
| Grassy vegetation | $0.97 - 0.98$ |
| Tree vegetation | $0.96 - 0.97$ |
| Snow (old) | 0.97 |
| Snow (fresh) | 0.99 |

**Table 3:** Soil emissivity for surface types (Brutsaert, 2005).

Since surface soil temperature measurements are only available at a few measurement sites, if the difference
between soil and air temperatures is not too big, it is possible to simulate $L_\uparrow$ using the air temperature, Park
et al. (2008). In our approach three different types of temperature were used to simulate $L_\uparrow$, specifically: surface




soil temperature; air temperature at 2 m height; and soil temperature at 4 cm depth.
The LWRB package (see flowchart in figure1) is part of the JGrass-NewAge system and was preliminary
tested in Formetta et al. (2014b). Model inputs depend on the specific SM being implemented and the purpose
of the run being performed (calibration, verification, simulation). The inputs are meteorological observations
such as air temperature, relative humidity, incoming solar radiation, and sky clearness index. The LWRB is also
fed by other JGrass-NewAGE components, such as the shortwave radiation balance (SWRB) (Formetta et al.,
2013). To test model performances (i.e. verification), the LWRB can be connected to the system's Verification
component; to execute the parameter calibration algorithm (Formetta et al., 2014a), it can be connected to the
LUCA (Let Us CAlibrate) component. In turn, all these components can and/or need to be connected to other
ones, as the problem under examination may require.
Further information about the SMs used is available in table 1 and in Carmona et al. (2014).
Model outputs are $L_\downarrow$ and $L_\uparrow$. These can be provided in single points of specified coordinates or over a
whole geographic area, represented as a raster map. For the latter case a digital elevation model (DEM) of the
study area is necessary in input.

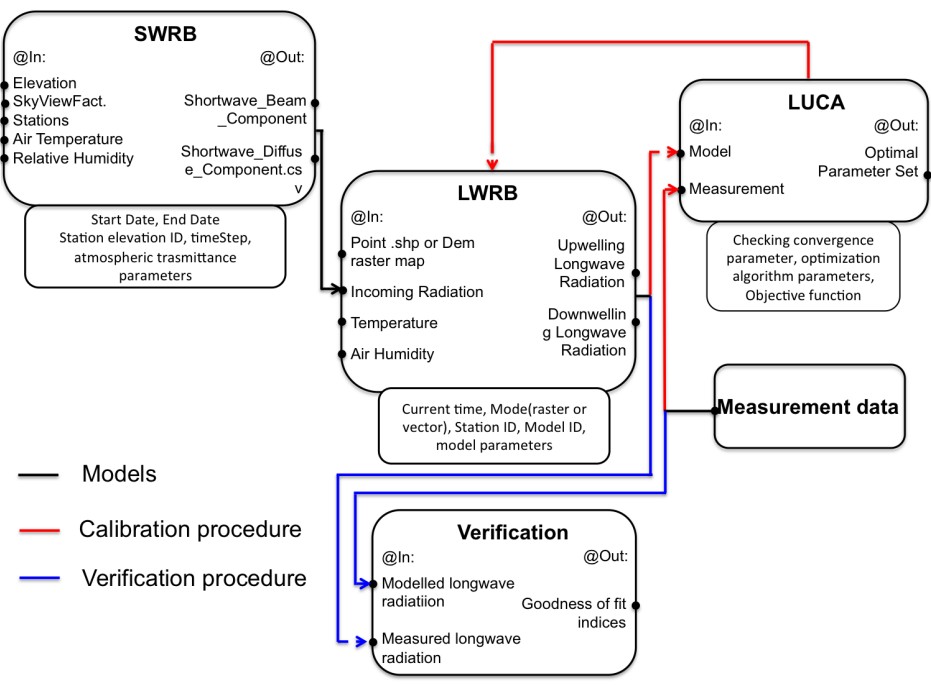

**Figure 1:** The LWRB component of JGrass-NewAge and the flowchart to model longwave radiation.

## 2.1 Calibration of $L_\downarrow$ longwave radiation models

Model calibration estimates the site-specific parameters of $L_\downarrow$ models by tweaking them with a specific algorithm
in order to best fit measured data. To this end, we use the LUCA calibration algorithm proposed in (Hay et al.,





2006), which is a part of the OMS core and is able to optimize parameters of any OMS component. LUCA is
a multiple-objective, stepwise, and automated procedure. As with any automatic calibration algorithm, it is
based on two elements: a global search algorithm; and the objective function(s) to evaluate model performance.
In this case, the global search algorithm is the Shuffled Complex Evolution, which has been widely used and
described in literature (e.g., Duan et al., 1993). As the objective function we use the Kling-Gupta Efficiency
(KGE), which is described below, but LUCA could use other objective functions just as well.
The calibration procedure for $L_\downarrow$ follows these steps:
• The theoretical solar radiation at the top of the atmosphere ($I_{top}$) is computed using the SWRB (see
Figure 1);
• The clearness index, $c$, is calculated as the ratio between the measured incoming solar radiation ($I_m$) and
$I_{top}$;
• Clear-sky and cloud-cover hours are detected by a threshold on the clearness index (equal to 0.6), providing
two subsets of measured $L_\downarrow$, which are $L_{\downarrow clear}$ and $L_{\downarrow cloud}$;
• The parameters X, Y, and Z for the models in table 1 are optimised using the subset $L_{\downarrow clear}$ and setting
$a$=0 in eq. 3.
• The parameters $a$ and $b$ for eq. 3 are optimized using the subset $L_{\downarrow cloud}$ and using the X, Y, and Z values
computed in the previous step.
The calibration procedure provides the optimal set of parameters at a given location for each of the ten
models.
As well as parameter calibration, we carry out a model parameter sensitivity analysis and we provide a linear
regression model relating a set of site-specific optimal parameters with easily available climatic variables, such
as mean air temperature, relative humidity, precipitation and altitude.

## 2.2 Verification of $L_\downarrow$ and $L_\uparrow$ longwave radiation models

As presented in previous applications (e.g. Hatfield et al. (1983), Flerchinger et al. (2009)), we use the SMs
with the original coefficients from literature (i.e. the parameters of table 2) and compare the performances of
the models against available measurements of $L_\downarrow$ and $L_\uparrow$ for each site. The goodness of fit is evaluated by using
two goodness-of-fit estimators: the Kling-Gupta Efficiency (KGE) presented in Gupta et al. (2009); and the
root mean square error (RMSE).
The KGE (eq. 4) is able to incorporate into one objective function three different statistical measures of
the relation between measured (M) and simulated (S) data: (i) the correlation coefficient, $r$ ; (ii) the variability
error, $a = \sigma_S/\sigma_M$; and (iii) the bias error, b=$\mu_S/\mu_M$. In these definitions $\mu_S$ and $\mu_M$ are the mean values,
while $\sigma_S$ and $\sigma_M$ are the standard deviations, of measured and simulated time series.

$$KGE = 1 - \sqrt{(r-1)^2 + (a-1)^2 + (b-1)^2} \qquad (4)$$




The RMSE, on the other hand, is presented in eq. 5:

$$RMSE = \sqrt{\frac{1}{N}\sum_{i=1}^{N}(M_i - S_i)^2} \qquad (5)$$

where M and S represents the measured and simulated data respectively.

## 3   The study area: the AmeriFlux Network

To test and calibrate the LWRB SMs we use twenty-four meteorological stations of the AmeriFlux Network
(http://ameriflux.ornl.gov). AmeriFlux is a network of sites that measure water, energy, and CO2 ecosystem
fluxes in North and South America. The dataset is widely known and used for biological and environmental
applications. To cite a few, Xiao et al. (2010) used Ameriflux data in a study on gross primary production data,
Kelliher et al. (2004) in a study on carbon mineralization, and Barr et al. (2012) in a study on hurricanes. Data
used in this study are the Level 2, 30-minute average data. Complete descriptions and downloads are available
at the Web interface located at http://public.ornl.gov/ameriflux/.
We have chosen twenty-four sites that are representative of most of the USA and span a wide climatic range:
going from the arid climate of Arizona, where the average air temperature is 16 °C and the annual precipitation
is 350 mm, to the equatorial climate of Florida, where the average air temperature is 24 °C and the annual
precipitation is 950 mm. Some general and climatic characteristics for each site are summarized in table 4, while
figure 2 shows their locations. The 30-minute average data have been cumulated to obtain continuous time series
of averaged, hourly data for longwave radiation, air and soil temperature, relative humidity, precipitation, and
soil water content.

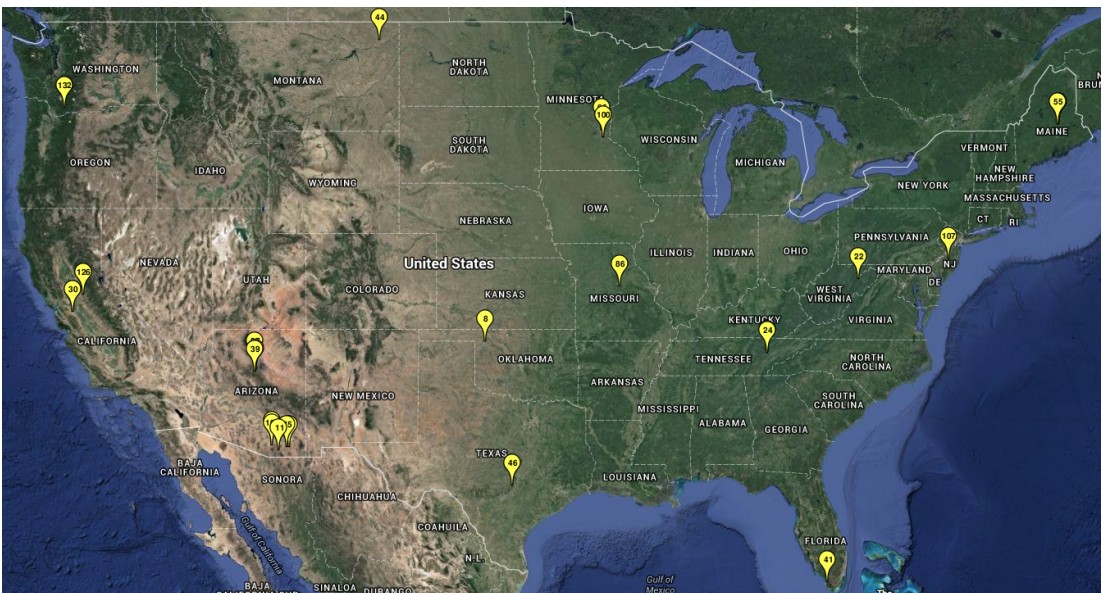

**Figure 2:**  Test site locations in the United State of America.





| SiteID | State | Latitude | Longitude | Elevation (m) | Climate | T ($^oC$) | Data period |
|---|---|---|---|---|---|---|---|
| 1 | AZ | 31.908 | −110.840 | 991 | semiarid | 19 | 2008 − 2013 |
| 2 | AZ | 31.591 | −110.509 | 1469 | temperate,arid | 16 | 2002 − 2011 |
| 3 | AZ | 31.744 | −110.052 | 1372 | temperate,semi-arid | 17 | 2007 − 2013 |
| 4 | AZ | 31.737 | −109.942 | 1531 | temperate,semi-arid | 17 | 2004 − 2013 |
| 5 | AZ | 31.821 | −110.866 | 116 | subtropical | 19 | 2004 − 2014 |
| 6 | AZ | 35.445 | −111.772 | 2270 | warm temperate | 9 | 2005 − 2010 |
| 7 | AZ | 35.143 | −111.727 | 2160 | warm temperate | 9 | 2005 − 2010 |
| 8 | AZ | 35.089 | −111.762 | 2180 | warm temperate | 8 | 2005 − 2010 |
| 9 | CA | 37.677 | −121.530 | 323 | mild | 16 | 2010 − 2012 |
| 10 | CA | 38.407 | −120.951 | 129 | mediterranean | 15 | 2000 − 2012 |
| 11 | FL | 25.365 | −81.078 | 0 | equatorial savannah | 24 | 2004 − 2011 |
| 12 | ME | 45.207 | −68.725 | 61 | temperate continental | 5 | 1996 − 2008 |
| 13 | ME | 45.204 | −68.740 | 60 | temperate continental | 6 | 1996 − 2009 |
| 14 | MN | 44.995 | −93.186 | 301 | continental | 6 | 2005 − 2009 |
| 15 | MN | 44.714 | −93.090 | 260 | snowy, humid summer | 8 | 2003 − 2012 |
| 16 | MO | 38.744 | −92.200 | 219 | temperate continental | 13 | 2004 − 2013 |
| 17 | MT | 48.308 | −105.102 | 634 | continental | 5 | 2000 − 2008 |
| 18 | NJ | 39.914 | −74.596 | 30 | temperate | 12 | 2005 − 2012 |
| 19 | OK | 36.427 | −99.420 | 611 | cool temperate | 15 | 2009 − 2012 |
| 20 | TN | 35.931 | −84.332 | 286 | temperate continental | 15 | 2005 − 2011 |
| 21 | TN | 35.959 | −84.287 | 343 | temperate | 14 | 1994 − 2007 |
| 22 | TX | 29.940 | −97.990 | 232 | warm temperate | 20 | 2004 − 2012 |
| 23 | WA | 45.821 | −121.952 | 371 | strongly seasonal | 9 | 1998 − 2013 |
| 24 | WV | 39.063 | −79.421 | 994 | temperate | 7 | 2004 − 2010 |

**Table 4:** Some general and climatic characteristics of the sites used for calibration: elevation is the site elevation above sea level, T is the annual average temperature, and data period refers to the period of available measurements.

## 4 Results

### 4.1 Verification of $L_\downarrow$ models with literature parameters

When implementing the ten $L_\downarrow$ SMs using the literature parameters, in many cases, they show a strong bias in reproducing measured data. A selection of representative cases is presented in Figure 3, which shows scatterplots for four SMs in relation to one measurement station. The black points represent the hourly estimates of $L_\downarrow$ provided by literature formulations, while the solid red line represents the line of optimal predictions. Model 1 (Ångström (1915)) shows a tendency to lie below the 45 degree line, indicating a negative bias (percent bias of -9.8) and, therefore, an underestimation of $L_\downarrow$. In contrast, model 9 ( Prata (1996)) shows an overestimation of $L_\downarrow$ with a percent bias value of 26.3.

Figure 4 presents the KGE (first column) and RMSE (second column) obtained for each model under clear-sky conditions, grouped by classes of latitude and longitude. Model 8 (Konzelmann et al. (1994)) does not perform very well for some reason. Its KGE values range between 0.16 and 0.41, while its RMSE values are higher than 100 $W/m^2$, with a maximum of 200 $W/m^2$. Model 6 (Idso (1981)) and model 2 (Brunt (1932)) provide the best results, independently of the latitude and longitude ranges where they are applied. Their KGE values are between 0.75 and 0.94, while the RMSE has a maximum value of 39 $W/m^2$.



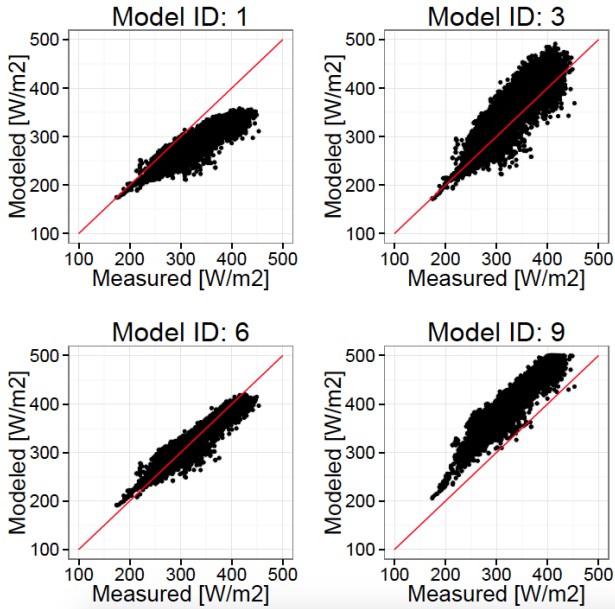

**Figure 3:** Results of the clear-sky simulation for four literature models using data from Howland Forest (Maine).

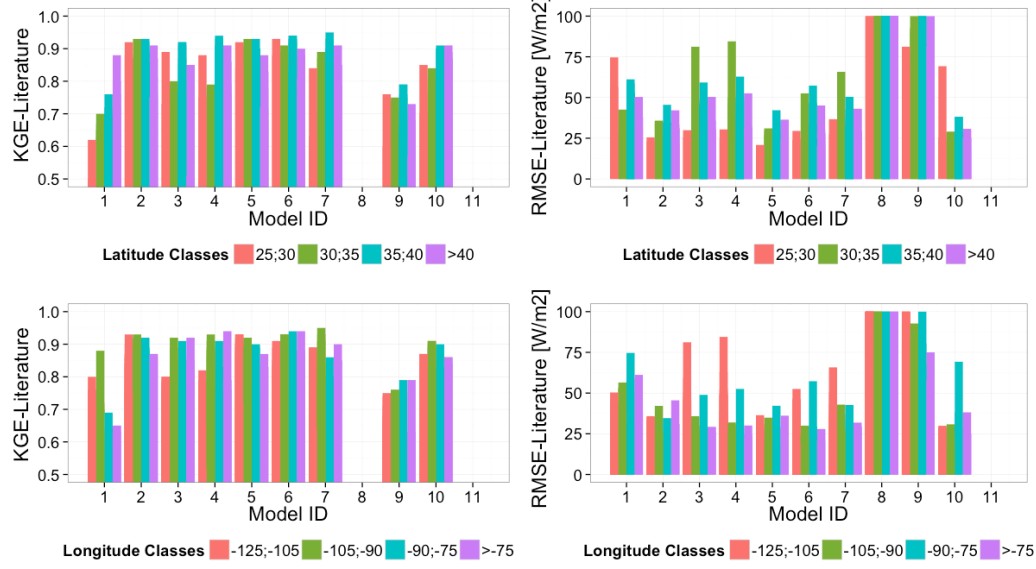

**Figure 4:** KGE and RMSE values for each clear-sky simulation using literature formulations, grouped by classes of latitude and longitude. The values of the KGE shown are those above 0.5: in this case, model 8 KGE values are not represented as they are between 0.16 and 0.41. The range of RMSE is 0-100 $W/m^2$.





## 4.2 $L_\downarrow$ models with site-specific parameters

The calibration procedure greatly improves the performances of all ten SMs. Optimized model parameters for each model are reported in the supplementary material. Figure 5 presents the KGE and RMSE values for clear-sky conditions grouped by classes of latitude and longitude. The percentage of KGE improvement ranges from its maximum value of 80% for model 8 (which is not, however, representative of the mean behavior of the SMs) to less than 10% for model 6, with an average improvement of around 35%. Even though variations in model performances with longitude and latitude classes still exist when using optimized model parameters, the magnitude of these variations is reduced with respect to the use of literature formulations. The calibration procedure reduces the RMSE values for all the models to below 50 $W/m^2$, with the exception of model 8, which now has a maximum of 58 $W/m^2$.

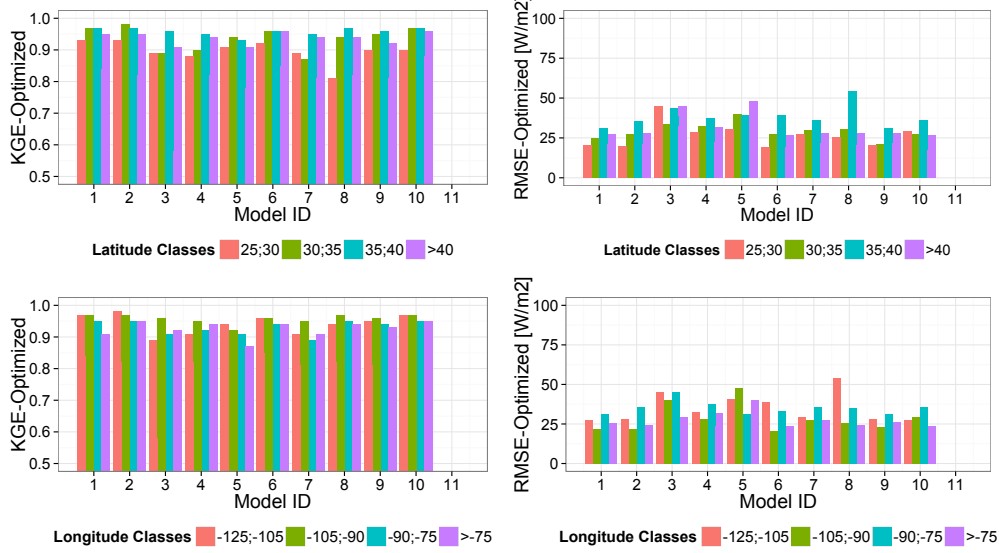

**Figure 5:** KGE (best is 1) and RMSE (best is 0) values for each optimized formulation in clear-sky conditions, grouped by classes of latitude and longitude. Only values of KGE above 0.5 are shown.

Figure 6 presents KGE and RMSE values for each model under all-sky conditions, grouped by latitude and longitude classes. In general, for all-sky conditions we observe a deterioration of KGE and RMSE values with respect to the clear-sky optimized case, with a decrease in KGE values up to a maximum of 25% for model 10. This may be due to uncertainty incorporated in the formulation of the cloudy-sky correction model (eq. 3): it seems that sometimes the cloud effects are not accounted for appropriately. This, however, is in line with the findings of Carmona et al. (2014).





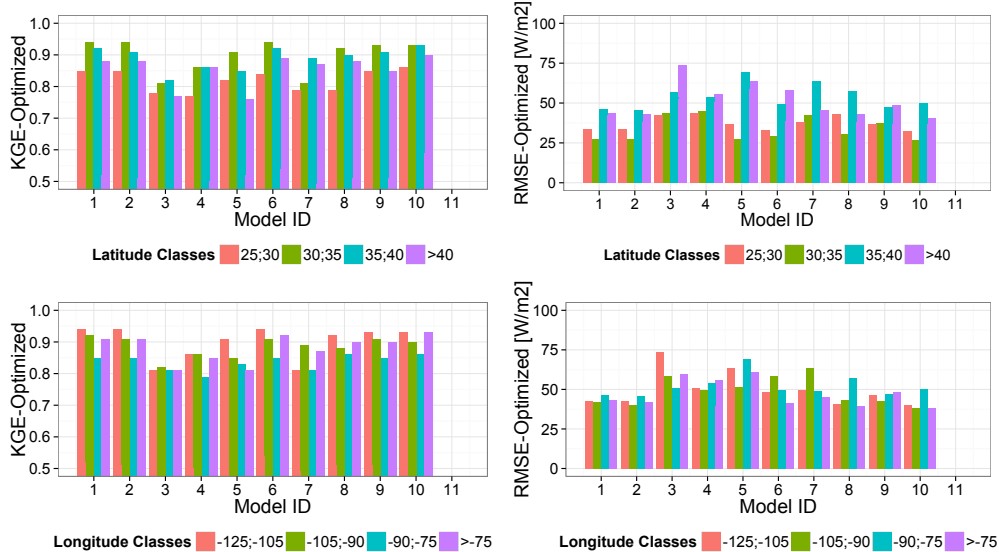

**Figure 6:** KGE and RMSE values for each model in all-sky conditions, grouped by classes of latitude and longitude. Only values of KGE above 0.5 are shown.

## 4.3 Sensitivity analysis of $L_\downarrow$ models

For each $L_\downarrow$ model we carry out a model parameters sensitivity analysis to investigate the effects and significance of parameters on performance for different model structures (i.e. models with one, two, and three parameters). The analyses are structured according to the following steps:

- we start with the optimal parameter set, computed by the optimization process for the selected model;

- all parameters are kept constant and equal to the optimal parameter set, except for the parameter under analysis;

- 1000 random values of the analyzed parameter are picked from a uniform distribution centered on the optimal value with width equal to ± 30% of the optimal value; in this way 1000 model parameter sets were defined and 1000 model runs were performed;

- 1000 values of KGE are computed by comparing the model outputs with measured time series.

The procedure was repeated for each parameter of each model. Figures 7-a and 7-b summarize the sensitivity analysis results for models 1 to 5 and models 6 to 10, respectively. Each figure presents three columns, one for each parameter. Considering model 1 and parameter X: the range of X is subdivided into ten equal-sized classes and for each class the corresponding KGE values are presented as a boxplot. A smooth blue line passing through the boxplot medians is added to highlight any possible pattern to parameter sensitivity. A flat line indicates that the model is not sensitive to parameter variation about optimal value. Results suggest that models with one and two parameters are all sensitive to parameter variation, presenting a peak in KGE in correspondence with their optimal values; this is more evident in models with two parameters. Models with three parameters tend





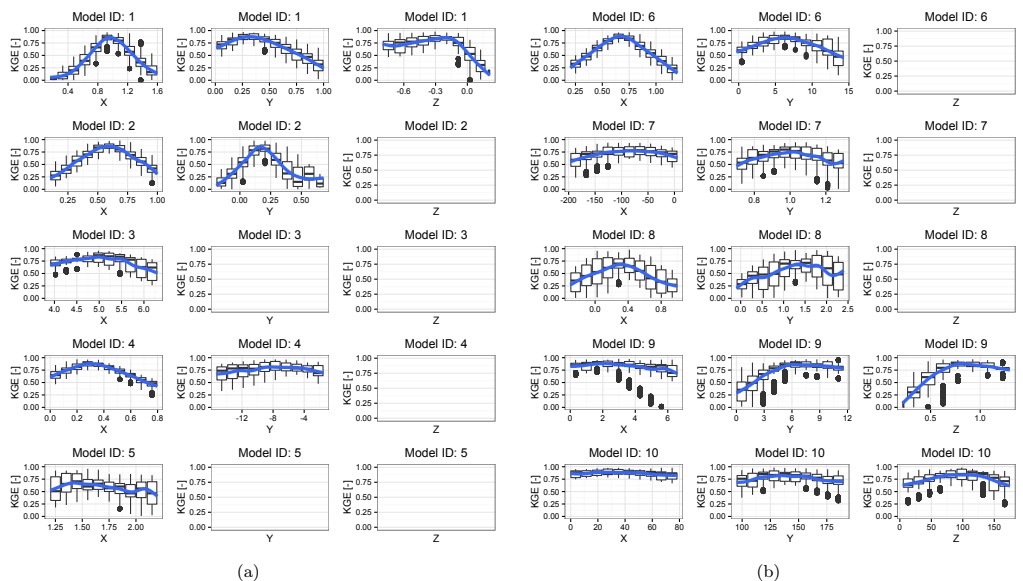

**Figure 7:** Results o the model parameters sensitivity analysis.

to have at least one insensitive parameter, except for model 1, that could reveal a possible overparameterization of the modeling process.

## 4.4 Regression model for parameters of $L_\downarrow$ models

The calibration procedure that allows the estimation of site specific parameters for $L_\downarrow$ models requires measured downwelling longwave data. Because these measurements are rarely available, we implement a straightforward multivariate linear regression (Chambers et al., 1992; Wilkinson and Rogers, 1973) to relate the site-specific parameters X, Y and Z to a set of easily available site specific climatic variables, used as regressors $r_i$. To perform the regression we use the open-source R software (https://cran.r-project.org) and to select the best regressors we use algorithms known as "best subsets regression", which are available in all common statistical software packages. The script containing the regression model is available, with the complementary material, at the web page of this paper: http://abouthydrology.blogspot.it/2015/07/site-specific-long-wave-radiation.html.

The regressors we have selected are: mean annual air temperature, relative humidity, precipitation, and altitude. The models that we use for the three parameters are presented in equations (6), (7), and (8):

$$X = a_X + \sum_{k=1}^{N} \alpha_k \cdot r_k + \epsilon_X \qquad (6)$$

$$Y = a_Y + \sum_{k=1}^{N} \beta_k \cdot r_k + \epsilon_Y \qquad (7)$$



$$Z = a_Z + \sum_{k=1}^{N} \gamma_k \cdot r_k + \epsilon_Z \tag{8}$$

where N=4 is the number of regressors (annual mean air temperature, relative humidity, precipitation, and altitude); $r_k$ with k=1,.., 4 are the regressors; $a_X$, $a_Y$, and $a_Z$ are the intercepts; $\alpha_k$, $\beta_k$, and $\gamma_k$ are the coefficients; and $\epsilon_X$, $\epsilon_Y$, and $\epsilon_Z$ are the normally distributed errors. Once the regression parameters are determined, the end-user can estimate site specific X, Y and Z parameter values for any location by simply substituting the values of the regressors in the model formulations.

The performances of the $L_\downarrow$ models using parameters assessed by linear regression are evaluated through the leave-one-out cross validation (Efron and Efron, 1982). We use 23 stations as training-sets for equations (6), (7), and (8) and we perform the model verification on the remaining station. The procedure is repeated for each of the 24 stations.

The cross validation results for all $L_\downarrow$ models and for all stations are presented in figures (8) and (10),grouped by classes of latitude and longitude, respectively. They report the KGE comparison between the $L_\downarrow$ models with their original parameters (in red) and with the regression model parameters (in blue).

In general, the use of parameters estimated with regression model gives a good estimation of $L_\downarrow$, with KGE values of up to 0.97. With respect to the classic formulation, model performance with regression parameters improved for all the models, in particular for model 8 in which the KGE improved from a minimum of 0.16 for the classic formulation to a maximum of 0.97.





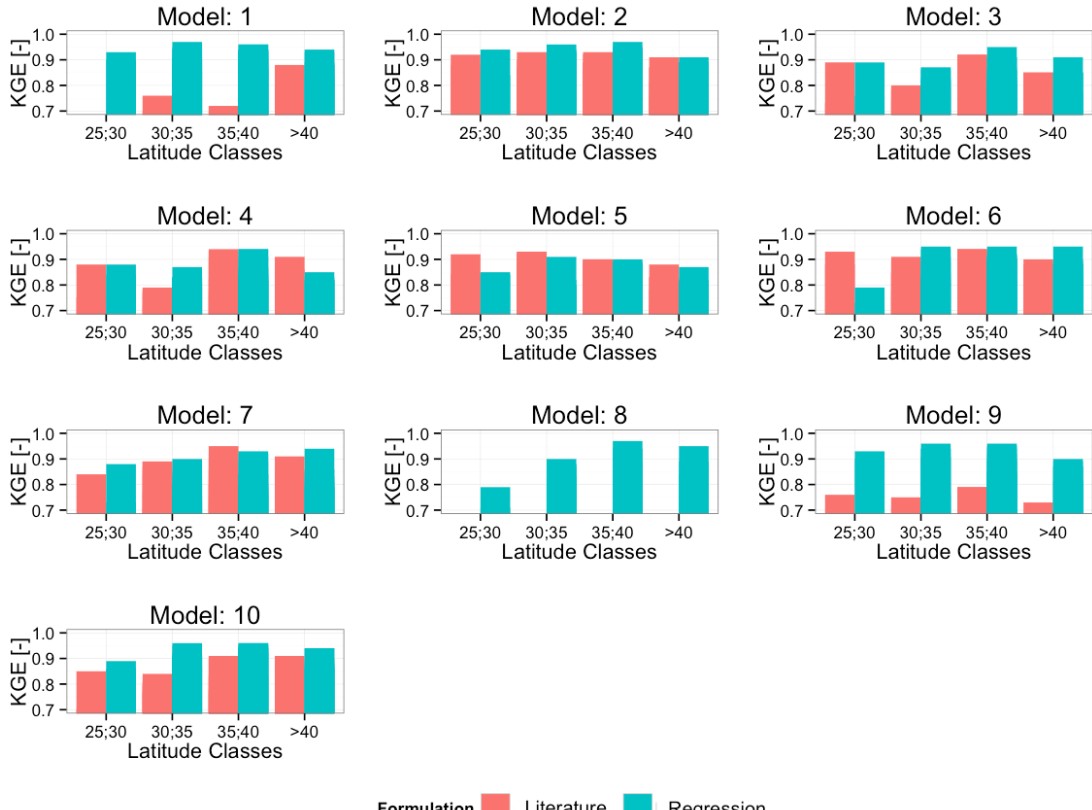

**Figure 8:** Comparison between model performances obtained with regression and classic parameters: the KGE values shown are those above 0.7 and results are grouped by latitude classes.



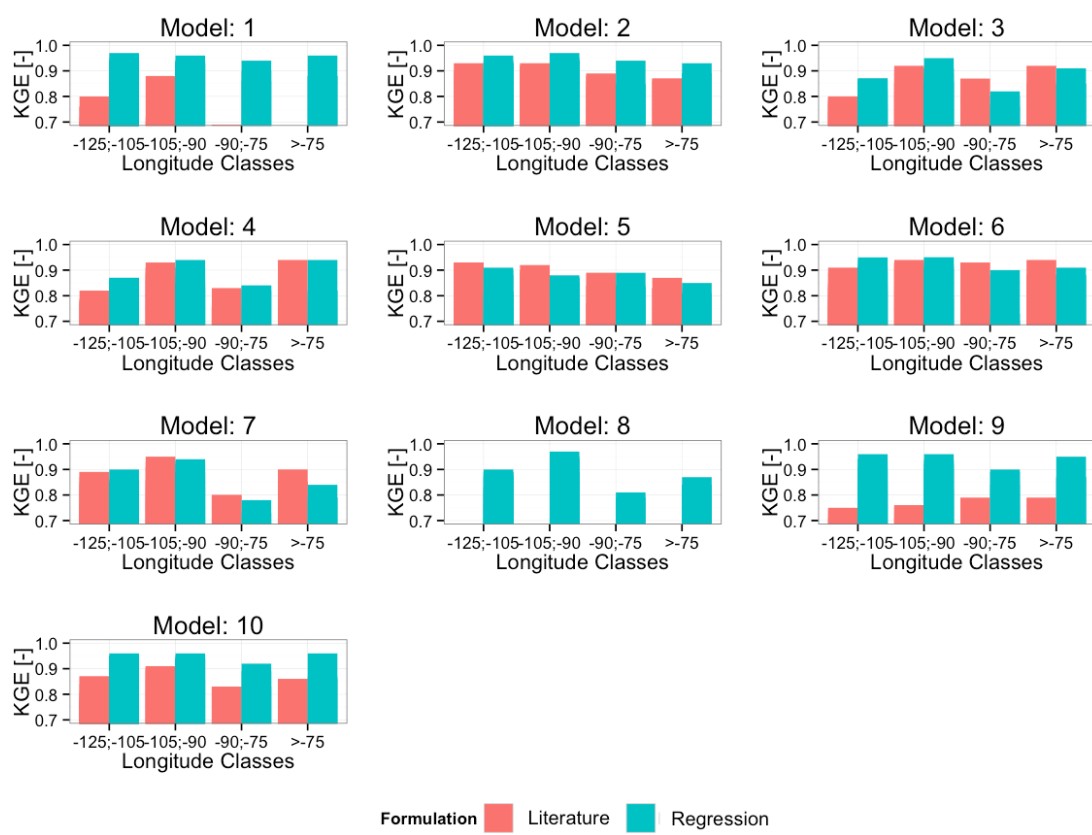

**Figure 9:** Comparison between model performances obtained with regression and classic parameters: the KGE values shown are those above 0.7 and results are grouped by longitude classes.





## 4.5 Verification of the $L_\uparrow$ model

Figure 10 presents the results of the $L_\uparrow$ simulations obtained using the three different temperatures available at experimental sites: soil surface temperature (skin temperature), air temperature, and soil temperature (measured at 4 cm below the surface). The figure shows the performances of the $L_\uparrow$ model for the three different temperatures used in terms of KGE, grouping all the stations for the whole simulation period according to season. This highlights the different behaviors of the model for periods where the differences in the three temperatures are larger (winter) or negligible (summer). The values of soil emissivity are assigned according the soil surface type, according to Table 4 (Brutsaert, 2005).

The best fit between measured and simulated $L_\uparrow$ is obtained with the surface soil temperature, with an all-season average KGE of 0.80. Unfortunately, the soil surface temperature is not an easily available measurement. In fact, it is available only for 8 sites of the 24 in the study area. Very good results are also obtained using the air temperature, where the all-season average KGE is around 0.76. The results using air temperature present much more variance compared to those obtained with the soil surface temperature. However, air temperature (at 2 m height) is readily available measure, in fact it is available for all 24 sites.

The use soil temperature at 4 cm depth provides the least accurate results for our simulations, with an all-season average KGE of 0.46. In particular, the use of soil temperature at 4 cm depth during the winter is not able to capture the dynamics of $L_\uparrow$. It does, however, show a better fit during the other seasons. This could be because during the winter there is a substantial difference between the soil and skin temperatures, as also suggested in Park et al. (2008).




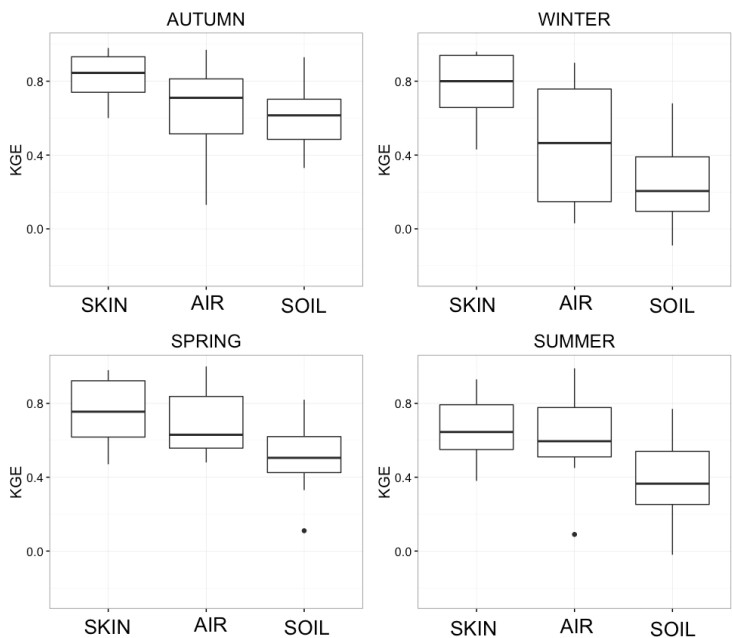

**Figure 10:** Boxplots of the KGE values obtained by comparing modeled upwelling longwave radiation, computed with different temperatures (soil surface temperature (SKIN), air temperature (AIR), and soil temperature (SOIL)), against measured data. Results are grouped by seasons.

## 5   Conclusions

This paper presents the LWRB package, a new modeling component integrated into the JGrass-NewAge system to model upwelling and downwelling longwave radiation. It includes ten parameterizations for the computation of $L_\downarrow$ longwave radiation and one for $L_\uparrow$. The package uses all the features offered by the JGrass-NewAge system, such as algorithms to estimate model parameters and tools for managing and visualizing data in GIS.

The LWRB is tested against measured $L_\downarrow$ and $L_\uparrow$ data from twenty-four AmeriFlux test-sites located all over continental USA. The application for $L_\downarrow$ longwave radiation involves model parameter calibration, model performance assessment, and parameters sensitivity analysis. Furthermore, we provide a regression model that estimates optimal parameter sets on the basis of local climatic variables, such as mean annual air temperature, relative humidity, and precipitation. The application for $L_\uparrow$ longwave radiation includes the evaluation of model performance using three different temperatures.

The main achievements of this work include: i) a broad assessment of the classic $L_\downarrow$ longwave radiation parameterizations, which clearly shows that the Idso (1981) and Brunt (1932) models are the more robust and reliable for all the test sites, confirming previous results; ii) a site specific assessment of the $L_\downarrow$ longwave radiation model parameters for twenty-four AmeriFlux sites that improved the performances of all the models; iii) the set up of a regression model that provides an estimate of optimal parameter sets on the basis climatic data; iv) an assessment of $L_\uparrow$ model performances for different temperatures (skin temperature, air temperature, and soil temperature at 4 cm below surface), which shows that the skin and the air temperature are better





proxy for the $L_\uparrow$ longwave radiation.
The integration of the package into JGrass-NewAge will allow users to build complex modeling solutions
for various hydrological scopes. In fact, future work will include the link of the LWRB package to the existing
components of JGrass-NewAge to investigate $L_\downarrow$ and $L_\uparrow$ effects on evapotranspiration, snow melting, and glacier
evolution.

# ACKNOWLEDGEMENTS

The authors are grateful to the AmeriFlux research community for providing the high-quality public data sets.
In particular, we want to thank the principal investigators of each site: Shirley Kurc Papuga (AZ), Tilden
P. Meyers (AZ), Russ Scott (AZ), Tom Kolb (AZ), Sonia Wharton (CA), Dennis D. Baldocchi (CA), Jordan
G.Barr (FL), Vic C. Engel (FL), Jose D. Fuentes (FL), Joseph C. Zieman (FL), David Y. Hollinger (ME), Joe
McFadden (MN), John M. Baker (MN), Timothy J. Griffis (MN), Lianhong Gu (MO), Kenneth L. Clark (NJ),
Dave Billesbach (OK), James A. Bradford (OK), Margaret S. Torn (OK), James L. Heilman (TX), Ken Bible
(WA), Sonia Wharton (WA). The authors thank the CLIMAWARE Project, of the University of Trento (Italy),
and the GLOBAQUA Project, which have supported their research.

# Replicable Research

In order that interested researchers may replicate or extend our results, our codes are made available at
$https://github.com/geoframecomponents.$
Instructions for using the code can be found at:
$http://geoframe.blogspot.co.uk/2016/04/lwrb-component-latest-documentation.html.$
Regression of parameters were performed in R and are available at
$https://github.com/GEOframeOMSProjects/OMS\_Project\_LWRB/blob/master/docs/Regression.R$

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
