# Peer review of "Performances of site specific parameterizations of longwave"

_Hydrology and Earth System Sciences, 2016_

## Referee Comment (RC1) · Anonymous Referee #1 · 20 Jun 2016

The study evaluates the performance of site specific parameterizations of longwave radiation. Similar evaluations have already been done by other authors. What's special about this study are two points: 1) The model parameters have been randomly perturbed to analyze their sensitivity. 2) The site specific model parameters were also estimated with the help of multiple regressions against commonly available local and climatic variables. The results are interesting and definitely worth to be published in HESS after the following comments have been addressed:

Section 2: Please describe how the last and first hour of daylight was defined.

Section 4: There is hardly any discussion of the results. I suggest adding the discussion of the findings in the Result section.

Section 4.2: I miss a figure or table, which shows the variability of the site-specific

model parameters for the different stations analyzed. This information is necessary in order to judge the sensitivity of the parameters on the different climates. Possibly this is reported in the mentioned supplementary material, which I could not find!

Section 4.3: You write "you start with optimal parameter set". Is done for every station? Moreover, it might be worth mentioning that the all three parameters of model 10 seem to be quite robust.

Section 4.4: This section is really innovative and therefore its potential needs to be explored more. IN practice you often don't have stations nearby, which can be used as a training set. I would like to see how a Ameriflux station in northern Alaska (Arctic) and South America (Tropics) performs with your currently used training set. Is there a specific reason you don't show the RMSE for this section? Which models perform best in this section? As I understand the red bars in Figure 8 represent the same KGE values as the bars in Figure 4. A visual test with model 1 shows a disagreement for latitude class 30;35 and 35;40! Please explain.

Section 4.5: Did take into account the soil was snow covered for some time at some stations. Please discuss the effect of snow an your approach and how it influences your results?

Section 5:In the Conclusion section, I miss a focus on the actual results, i.e. the evaluation of the different site-specific parameterizations methods and the performance of the different models. For example, it is not enough to write "A broad assessment of the classic longwave radiation parameterizations clearly shows that the Idso (1981) and Brunt (1932) models are the more robust and reliable for all the test sites, confirming previous results". First, I don't "see" this. Please add information based on RMSE or KGE (however this should not be done in the Conclusion section). Second, add the references, which seem to confirm your results.

Minor Comments:

[Figure]

L1: Performance of site specific parameterizations. . ..

L15: for L in SMs

L29: I guess data also!

L44: water vapor deficit

L46: Be consistent - when using L you don't need to add downwelling or upwelling radiation.

L49: Instead of old references I suggest to replace it with newer ones, like doi:10.1007/s00704-012-0675-1 and doi:10.1016/j.coldregions.2013.12.004

L53-54: Why show the results only for this study?

L77: Delete "near surface" or replace with "screen level".

Table 1: The Monteith and Unsworth (1990) is missing in the Reference section, but I guess you mean Unsworth and Monteith (1975) anyway.

L103-105: Please reformulate. I suggest to make two sentences.

Figure 1: "incoming Radiation" in the LWRB box is confusing. Please replace with "Incoming Shortwave Radation".

L134: Why 0.6. Did you also test other thresholds?

L164: Could you please add some information about the used longwave instruments its measurement uncertainties.

L182-183: The reason is that the Konzelmann model was calibrated for the Greenland ice sheet, which has a totally different climate than you stations.

L225: For better understanding please link this part to the former section by changing the first sentence to: The just performed calibration procedure to estimate. . . requires. . .

L232: The URL is invalid: I suggest to add this information also to the supplementary material.

L244: figures (8) and (9)

---

## Author Comment (AC1) · 21 Jun 2016

| Model: 1 | | | |
|---|---|---|---|
| STATION ID | X | Y | Z |
| 8 | 0.93 | 0.18 | -0.07 |
| 11 | 1.03 | 0.50 | -0.21 |
| 22 | 0.90 | 0.21 | -0.23 |
| 24 | 0.93 | 0.54 | -0.41 |
| 30 | 0.90 | 0.27 | -0.34 |
| 37 | 0.99 | 0.36 | -0.17 |
| 38 | 0.93 | 0.29 | -0.41 |
| 39 | 1.01 | 0.46 | -0.33 |
| 41 | 1.00 | 0.40 | -0.13 |
| 44 | 1.01 | 0.27 | -0.14 |
| 46 | 0.95 | 0.38 | -0.28 |
| 54 | 0.98 | 0.44 | -0.36 |
| 55 | 1.17 | 0.48 | -0.14 |
| 62 | 1.00 | 0.33 | -0.23 |
| 66 | 1.00 | 0.28 | -0.17 |
| 75 | 0.96 | 0.38 | -0.22 |
| 86 | 0.93 | 0.31 | -0.40 |
| 100 | 1.06 | 0.55 | -0.27 |
| 101 | 1.08 | 0.43 | -0.14 |
| 102 | 1.16 | 0.36 | -0.08 |
| 107 | 0.90 | 0.26 | -0.13 |
| 126 | 0.97 | 0.32 | -0.23 |
| 129 | 0.90 | 0.27 | -0.26 |
| 132 | 0.89 | 0.17 | -0.38 |

| Model: 2 | | |
| --- | --- | --- |
| STATION ID | X | Y |
| 8 | 0.65 | 0.14 |
| 11 | 0.44 | 0.28 |
| 22 | 0.57 | 0.16 |
| 24 | 0.58 | 0.18 |
| 30 | 0.48 | 0.27 |
| 37 | 0.49 | 0.31 |
| 38 | 0.47 | 0.32 |
| 39 | 0.43 | 0.33 |
| 41 | 0.52 | 0.22 |
| 44 | 0.60 | 0.10 |
| 46 | 0.62 | 0.16 |
| 54 | 0.61 | 0.17 |
| 55 | 0.61 | 0.17 |
| 62 | 0.53 | 0.23 |
| 66 | 0.66 | 0.11 |
| 75 | 0.61 | 0.20 |
| 86 | 0.59 | 0.17 |
| 100 | 0.72 | 0.10 |
| 101 | 0.69 | 0.13 |
| 102 | 0.59 | 0.21 |
| 107 | 0.71 | 0.13 |
| 126 | 0.62 | 0.19 |
| 129 | 0.60 | 0.19 |
| 132 | 0.67 | 0.11 |

| Model: 3 | |
|---|---|
| STATION ID | X |
| 8 | 5.16 |
| 11 | 4.44 |
| 22 | 5.00 |
| 24 | 5.24 |
| 30 | 4.28 |
| 37 | 4.85 |
| 38 | 4.70 |
| 39 | 4.42 |
| 41 | 5.93 |
| 44 | 4.18 |
| 46 | 5.20 |
| 54 | 5.22 |
| 55 | 5.22 |
| 62 | 4.69 |
| 66 | 4.75 |
| 75 | 4.77 |
| 86 | 5.03 |
| 100 | 5.19 |
| 101 | 4.72 |
| 102 | 4.68 |
| 107 | 5.30 |
| 126 | 4.34 |
| 129 | 5.70 |
| 132 | 4.40 |

| Model: 4 | | |
|:---:|:---:|:---:|
| **STATION ID** | **X** | **Y** |
| 8 | 0.25 | -6.99 |
| 11 | 0.43 | -3.86 |
| 22 | 0.37 | -10.59 |
| 24 | 0.33 | -7.86 |
| 30 | 0.30 | -2.11 |
| 37 | 0.38 | -6.94 |
| 38 | 0.38 | -6.08 |
| 39 | 0.41 | -5.08 |
| 41 | 0.36 | -13.19 |
| 44 | 0.33 | -2.83 |
| 46 | 0.30 | -8.00 |
| 54 | 0.30 | -14.85 |
| 55 | 0.30 | -14.81 |
| 62 | 0.39 | -4.74 |
| 66 | 0.32 | -6.47 |
| 75 | 0.28 | -4.01 |
| 86 | 0.31 | -6.37 |
| 100 | 0.21 | -8.14 |
| 101 | 0.24 | -3.43 |
| 102 | 0.29 | -3.58 |
| 107 | 0.22 | -8.53 |
| 126 | 0.25 | -1.90 |
| 129 | 0.28 | -14.98 |
| 132 | 0.24 | -1.81 |

| Model: 5 | | |
| --- | --- | --- |
| STATION ID | X | Y |
| 8 | 1.62 | 11.50 |
| 11 | 1.69 | 9.84 |
| 22 | 1.52 | 6.45 |
| 24 | 1.64 | 5.45 |
| 30 | 1.74 | 9.98 |
| 37 | 1.77 | 2.39 |
| 38 | 1.77 | 6.84 |
| 39 | 1.72 | 6.38 |
| 41 | 1.83 | 4.52 |
| 44 | 1.49 | 10.36 |
| 46 | 1.75 | 7.00 |
| 54 | 1.58 | 5.39 |
| 55 | 1.58 | 8.47 |
| 62 | 1.67 | 4.71 |
| 66 | 1.70 | 6.59 |
| 75 | 1.78 | 10.83 |
| 86 | 1.65 | 3.79 |
| 100 | 1.62 | 8.53 |
| 101 | 1.57 | 3.99 |
| 102 | 1.79 | 8.87 |
| 107 | 1.70 | 4.67 |
| 126 | 1.84 | 8.40 |
| 129 | 1.72 | 2.82 |
| 132 | 1.59 | 8.95 |

| Model: 6 | | |
|---|---|---|
| STATION ID | X | Y |
| 8 | 0.70 | 4.35 |
| 11 | 0.54 | 12.08 |
| 22 | 0.65 | 3.24 |
| 24 | 0.63 | 6.58 |
| 30 | 0.60 | 12.17 |
| 37 | 0.59 | 14.00 |
| 38 | 0.58 | 14.00 |
| 39 | 0.55 | 14.00 |
| 41 | 0.67 | 5.40 |
| 44 | 0.63 | 4.65 |
| 46 | 0.68 | 5.50 |
| 54 | 0.65 | 7.05 |
| 55 | 0.65 | 7.04 |
| 62 | 0.60 | 9.96 |
| 66 | 0.71 | 2.56 |
| 75 | 0.68 | 8.36 |
| 86 | 0.65 | 6.09 |
| 100 | 0.75 | 3.43 |
| 101 | 0.72 | 6.16 |
| 102 | 0.66 | 9.89 |
| 107 | 0.75 | 5.08 |
| 126 | 0.84 | 0.03 |
| 129 | 0.67 | 6.48 |
| 132 | 0.70 | 4.77 |

| Model: 7 | | |
| --- | --- | --- |
| STATION ID | X | Y |
| 8 | -83.39 | 1.02 |
| 11 | -104.32 | 0.89 |
| 22 | -84.81 | 0.94 |
| 24 | -146.55 | 1.12 |
| 30 | -46.26 | 0.84 |
| 37 | -93.56 | 0.92 |
| 38 | -79.48 | 0.88 |
| 39 | -67.42 | 0.81 |
| 41 | -196.54 | 1.29 |
| 44 | -5.37 | 0.70 |
| 46 | -167.34 | 1.18 |
| 54 | -56.46 | 0.90 |
| 55 | -55.00 | 0.89 |
| 62 | -109.60 | 0.95 |
| 66 | -52.33 | 0.89 |
| 75 | -65.59 | 0.93 |
| 86 | -119.59 | 1.06 |
| 100 | -55.82 | 0.98 |
| 101 | -19.76 | 0.85 |
| 102 | -76.26 | 0.93 |
| 107 | -79.62 | 1.03 |
| 126 | 3.67 | 0.77 |
| 129 | -134.69 | 1.16 |
| 132 | 5.91 | 0.77 |

| Model: 8 | | |
|---|---|---|
| STATION ID | X | Y |
| 8 | 0.51 | 0.82 |
| 11 | 0.00 | 1.91 |
| 22 | 0.52 | 0.56 |
| 24 | 0.06 | 1.90 |
| 30 | 0.09 | 1.77 |
| 37 | 0.24 | 1.35 |
| 38 | 0.00 | 2.04 |
| 39 | 0.00 | 1.95 |
| 41 | 0.00 | 2.11 |
| 44 | 0.41 | 0.79 |
| 46 | 0.58 | 0.61 |
| 54 | 0.42 | 0.94 |
| 55 | 0.42 | 0.95 |
| 62 | 0.13 | 1.67 |
| 66 | 0.67 | 0.24 |
| 75 | 0.20 | 1.63 |
| 86 | 0.09 | 1.82 |
| 100 | 0.39 | 1.17 |
| 101 | 0.64 | 0.45 |
| 102 | 0.19 | 1.65 |
| 107 | 0.39 | 1.23 |
| 126 | 0.32 | 1.24 |
| 129 | 0.31 | 1.34 |
| 132 | 0.29 | 1.33 |

| Model: 9 | | | |
|---|---|---|---|
| STATION ID | X | Y | Z |
| 8 | 4.98 | 10.87 | 0.58 |
| 11 | 2.98 | 5.07 | 0.80 |
| 22 | 5.87 | 8.99 | 0.56 |
| 24 | 1.89 | 6.00 | 0.74 |
| 30 | 1.48 | 5.96 | 0.72 |
| 37 | 1.00 | 0.34 | 0.12 |
| 38 | 3.00 | 4.79 | 0.98 |
| 39 | 2.96 | 4.47 | 0.92 |
| 41 | 0.30 | 5.76 | 0.70 |
| 44 | 1.90 | 5.77 | 0.68 |
| 46 | 2.36 | 6.13 | 0.75 |
| 54 | 2.86 | 5.96 | 0.86 |
| 55 | 2.87 | 5.96 | 0.89 |
| 62 | 2.97 | 5.75 | 0.80 |
| 66 | 2.24 | 5.39 | 0.79 |
| 75 | 1.88 | 5.96 | 0.90 |
| 86 | 1.71 | 5.74 | 0.76 |
| 100 | 0.44 | 5.91 | 0.81 |
| 101 | 1.05 | 5.53 | 0.89 |
| 102 | 2.22 | 5.94 | 0.93 |
| 107 | 2.55 | 8.58 | 0.86 |
| 126 | 0.62 | 5.95 | 0.79 |
| 129 | 1.71 | 5.99 | 0.82 |
| 132 | 6.24 | 11.28 | 0.60 |

| Model: 10 | | | |
|---|---|---|---|
| STATION ID | X | Y | Z |
| 8 | 38.63 | 171.71 | 42.03 |
| 11 | 6.63 | 110.53 | 174.48 |
| 22 | 2.14 | 186.44 | 30.32 |
| 24 | 29.01 | 132.73 | 108.33 |
| 30 | 0.46 | 150.74 | 96.12 |
| 37 | 0.21 | 148.45 | 127.18 |
| 38 | 0.06 | 134.58 | 151.82 |
| 39 | 0.06 | 125.35 | 158.33 |
| 41 | 0.07 | 120.76 | 154.09 |
| 44 | 53.08 | 105.03 | 105.85 |
| 46 | 0.09 | 188.01 | 69.27 |
| 54 | 55.34 | 123.33 | 99.19 |
| 55 | 56.01 | 122.24 | 100.27 |
| 62 | 0.68 | 150.39 | 95.12 |
| 66 | 37.64 | 182.26 | 4.69 |
| 75 | 39.45 | 154.46 | 79.96 |
| 86 | 33.62 | 135.85 | 97.63 |
| 100 | 78.32 | 112.74 | 103.41 |
| 101 | 62.08 | 147.42 | 68.12 |
| 102 | 24.13 | 152.75 | 96.77 |
| 107 | 73.65 | 132.25 | 100.46 |
| 126 | 17.84 | 161.27 | 71.43 |
| 129 | 15.71 | 159.41 | 89.76 |
| 132 | 74.09 | 97.62 | 128.17 |

---

## Referee Comment (RC2) · Anonymous Referee #2 · 30 Jun 2016

**General comments:**

The study analyses the performance of 10 empirical parametrisations of incoming longwave radiation with original parameters, site-specific fitted parameters and parameters obtained from regression with average climate variables. The calibration and validation data is taken from the AmeriFlux network. Additionally, the study compares the accuracy of outgoing longwave radiation estimates using soil temperature, soil surface temperature and air temperature.

In most parts, the study repeats a similar analysis as other papers (Flerchinger et al., 2009; Juszak and Pellicciotti, 2013; Carmona et al., 2014), which is the comparison of parametrisations of incoming longwave radiation with original and fitted parameters.

The novelty of the study arises from the site-specific estimation of the parameters using multivariate linear regression. This part is interesting for future studies which do not have longwave radiation data available. As the multivariate linear regression is the new and relevant part of the study, Section 4.4 should be elaborated more and presented in more detail. If this part is emphasised strongly, the paper may be published in HESS after major revisions.

1. The results section mixes methods, results and discussion. The methods should be moved to the methods section, the discussion should be separate and longer to incorporate (i) Which models are best at all sites and when used with parameter estimates from the regression approach? (ii) Are the regressions likely to work outside the USA? (iii) What are possible sources of uncertainty?

2. Most formulas have either not been cited correctly (Table 1 of the manuscript), or the given empirical parameters (Table 2 of the manuscript) were derived for different units of the input variables and can thus not be used with other units and without adjustment. This is a serious issue as it affects the results and conclusions. It should be corrected and all graphs need to be updated. Also some of the conclusions like "Model 8 (Konzelmann et al. (1994)) does not perform very well for some reason." (Line 182) and "Regarding the $L \downarrow$ simulations, the Brunt (1932) and Idso (1981) SMs, in their literature formulations, provide the best performances in many of the sites." (Abstract) may be wrong.

3. Some of the cited literature does not appear in the references.

4. $c$ is used for the clearness index and the cloud cover fraction. Please rename one of them and write the equation to convert them.

5. State in more detail the results of the parameter estimate by regression and provide the formula for the best model including average parameters.

**Specific comments:**

**Abstract** The study described in the manuscript is largely independent of the hydrological model JGrass-NewAge. The authors do not present any results concerning hydrology. Thus, this model should not be central in the first sentences of the abstract.

**L13–15** These are really 3 points: (i) original formulation, (ii) site specific calibration, and (iii) parameter estimation based on average site characteristics

**L16** Name all variables instead of 'such as'

**L21–23** This conclusion may change with correct model formulation. The relative performance of the models should be discussed in more detail in a discussion section.

**L29** Remove this sentence.

**L31** $3 - 100\,\mu\,\mathrm{m}$ (1 is still shortwave radiation (Oke, 1987))

**L34** Remove 'very expensive', that is relative

**L58–59** I do not agree with this major advantage of the current study as compared to the former studies. The empirical formulations of longwave radiation are very simple equations that can be included easily in any model without the need of an open-source tool. Instead, the authors could refer to their parameter estimation approach: 'However, none of the above studies have developed a method to estimate the parameters for any location based on basic site characteristics and ready for practical use by other researchers and practitioners.' More sentences on the added value of this study are needed. What are the research questions?

**L68–74** Paragraph not needed

**L77** the 'k' of 'kg' should be lower-case; it would be more intuitive to provide the unit $W\,m^{-2}\,K^{-4}$ as $L$ is given in $W\,m^{-2}$

**eq. 3** It should be noted that this equation was proposed by Bolz (1949), and that there are other options that potentially work better (Flerchinger et al., 2009; Juszak and Pellicciotti, 2013). The authors should consider using Unsworth and Monteith (1975), which was recommended in both studies.

**L81** $c$ is not the clearness index but the cloud cover fraction (as in line 84)

**L87** Related how? Provide equation!

**Table 1** I have doubts that all formulas in Table 1 are correct and that the parameters in Table 2 have been adjusted to the units of water vapour pressure (and in some cases radiation). I suggest you check Juszak and Pelicciotti (2013) for adjusted parameters. More specifically please consider:

- Angstrom [1918] does not appear in the reference list. Please provide the correct reference and check the original publication or cite the paper you took the parameters from. Did you adjust the original parameters to match the units in which you computed the radiation and inserted humidity and temperature? I have doubts in the Angstrom case where one original publication computes the radiation in $cal\,cm^{-2}\,min^{-1}$ (Ångström, 1916). Ångström (1916) also uses $e^{Ze}$ instead of $10^{Ze}$.
- Brunt (1932) uses water vapour pressure in millibar not kPa. Did you adjust the parameter Y?
- Swinbank (1963) is clearly used wrongly. The parameters provided in Table 2 do not refer to the clear sky emissivity but to a formula that computes the radiation directly (without $\sigma \cdot T^4$), and in $mW\,cm^{-2}$.
- Brutsaert (1975) uses water vapour pressure in millibar not kPa. Please adjust the parameters X and Y.

- Monteith and Unsworth [1990] does not appear in the literature list. Please double-check the formula and parameters and provide the correct citation.
- Konzelmann et al. (1994) uses water vapour pressure in Pa not kPa. Please adjust the parameters X and Y.
- Dilley and O'Brien (1998) uses the given formula (Table 1) with the parameters (Table 2) to directly compute the longwave flux, not the emissivity. To get the emissivity, the formula has to be divided by $\sigma \cdot T^4$

Use round brackets for the reference year as in the rest of the manuscript.

**L116** No one-sentence paragraph, this sentence can be removed.

**Figure 1** How do $I_m$ and $I_{top}$ fit into this schematic? Only those variables are explained later in the text. The 'Modelled longwave radiation' and 'Measured longwave radiation' items in the Verification box are wrongly connected. Is the SWBR always modelled? Does that affect the optimisation process?

**L134** Did you try different thresholds? 0.6 seems quite low. Did you verify that you do not include cloudy or partly cloudy observations in the clear sky calibration? If you calibrate $\epsilon_{clear}$ at $c = 0.6$, $\epsilon_{all-sky}$ at that condition will be wrong as you compute it from $\epsilon_{all-sky} = \epsilon_{clear} \cdot (1 + a \cdot c^b)$ and $c \neq 0$.

**L143–144** Please provide all variables (not 'such as'); altitude is not a climatic variable.

**L156** What is $N$?

**L161–162** Sentence not relevant, remove it.

**L166–168** There is also a gradient towards the colder climate. Why did you choose these 24 stations and not all stations?

**Figure 2** Use same index for stations as in Table 4 and make the index bigger so it is
readable.

**Table 4** How was the climate defined? 'mild' and 'strongly seasonal' do not match the
classic categories.

**Section 4.1** Update section with correct model implementation and parameters.

**Figure 3** Name models in the caption

**Figures 4–6, 8–9** Use boxplots instead of barplots to show the variability within the
groups and the range of variation. Reorganise the content to have only two Fig-
ures: one for clear sky and one for all sky. In both figures, boxes for results of
(i) original parameters, (ii) fitted parameters, and (iii) parameters from regression
analysis should be next to each other to enable direct comparison. The figures
can be arranged in subplots either one per model, or one per latitude / longi-
tude class. Please choose colours that allow black+white printing and consider
colour-blind people.

**Section 4.2** Update section with correct model implementation and original parameters.

**L201–202** This should be moved and discussed in more detail in a discussion section.

**L213** Time series from which station? Was the analysis done for all stations?

**L206–214** This belongs to the methods section.

**Figure 7** Given the methods description, why is the peak not always in the middle of
the parameter range? Caption: 'of' is missing an 'f'; describe the meaning of the
boxes and the line!

**L225–243** This belongs to the methods section.

**Equation 8** Do not use 'a' as it is used for something else in Equation 4

**L244–250** Compare also with fitted parameters.

**Section 4.4** Update section with correct model implementation and original parameters.

**L267–269** This should be moved and discussed in more detail in a discussion section. How about snow cover? How about the different latitudes?

**Conclusions** Update section with correct model implementation and original parameters.

**Supplementary material** Please use the same station IDs as in the manuscript. Please include the detailed results of the parameter regression.

**Technical corrections:**

**L12** 24 instead of twenty-four

**L36** put references in brackets

**L40** put references in brackets

**L51** 'They' instead of 'It'

**L52** remove 'so'

**L64** put reference in brackets

**Table 1** caption: units not in italics

**L101** space missing before reference

**L104** put references in brackets

**L106** replace ';' with ','

**L122** remove brackets from reference

**L158** 24 instead of twenty-four

**L165** 24 instead of twenty-four

**L178** 1:1 instead of 45 degree

**L219** 'around the' instead of 'about'

**L231** 'supplementary' instead of 'complementary'

**L244** Figure 10 shows something else

**L275** 24 instead of twenty-four

**L284** 24 instead of twenty-four

**L303** Reformulate 'In order that'

**References**

Ångström, A.: A Study of the Radiation of the Atmosphere, Ph.D. thesis, Philosophical Faculty of Upsala, 1916.

Bolz, H. M.: Die Abhängigkeit der infraroten Gegenstrahlung von der Bewölkung, Zeitschrift für Meteorologie, 7, 201–203, 1949.

Brunt, D.: Notes on radiation in the atmosphere, Quarterly Journal of the Royal Meteorological Society, 58, 389–420, doi:10.1002/qj.49705824704, 1932.

Brutsaert, W.: On a Derivable Formula for Long-Wave Radiation From Clear Skies, Water Resources Research, 11, 742–744, doi:10.1029/WR011i005p00742, 1975.

Carmona, F., Rivas, R., and Caselles, V.: Estimation of daytime downward longwave radiation under clear and cloudy skies conditions over a sub-humid region, Theoretical and Applied Climatology, 115, 281–295, doi:10.1007/s00704-013-0891-3, 2014.

Dilley, A. C. and O'Brien, D. M.: Estimating downward clear sky long-wave irradiance at the surface from screen temperature and precipitable water, Quarterly Journal of the Royal Meteorological Society, 124, 1391–1401, doi:10.1002/qj.49712454903, 1998.

Flerchinger, G. N., Xaio, W., Marks, D., Sauer, T. J., and Yu, Q.: Comparison of algorithms for incoming atmospheric long-wave radiation, Water Resources Research, 45, W03 423, doi:10.1029/2008WR007394, 2009.

Juszak, I. and Pellicciotti, F.: A comparison of parameterizations of incoming longwave radiation over melting glaciers: Model robustness and seasonal variability, Journal of Geophysical Research: Atmospheres, 118, 3066–3084, doi:10.1002/jgrd.50277, 2013.

Konzelmann, T., van de Wal, R. S., Greuell, W., Bintanja, R., Henneken, E. A., and Abe-Ouchi, A.: Parameterization of global and longwave incoming radiation for the Greenland Ice Sheet, Global and Planetary Change, 9, 143–164, doi:10.1016/0921-8181(94)90013-2, 1994.

Oke, T. R.: Boundary layer climates, Methuen & Co, 2 edn., 1987.

Swinbank, W. C.: Long-wave radiation from clear skies, Quarterly Journal of the Royal Meteorological Society, 89, 339–348, doi:10.1002/qj.49708938105, 1963.

Unsworth, M. H. and Monteith, J. L.: Long-wave radiation at the ground 1. Angular distribution of incoming radiation, Quarterly Journal of the Royal Meteorological Society, 101, 13–24, doi:10.1002/qj.49710142703, 1975.

---

## Author Comment (AC2) · 12 Aug 2016

*The study evaluates the performance of site-specific parameterizations of longwave radiation. Similar evaluations have already been done by other authors. What's special about this study are two points: 1) The model parameters have been randomly perturbed to analyze their sensitivity. 2) The site-specific model parameters were also estimated with the help of multiple regressions against commonly available local and climatic variables. The results are interesting and definitely worth to be published in HESS after the following comments have been addressed:*

The authors thank the reviewer for the prompt revision and the interesting comments and suggestions he made. They definitely improved the quality of the paper. Below we replied one-to-one to each comment.

Q1) *Section 2: Please describe how the last and first hour of daylight was defined.*

A1) We thank the reviewer for the suggestion and we agree with it. We added the following sentence to specify how we computed the first and last hour of daylight.

"The computation of the first and last hour of the day are based on the model proposed in Formetta et al., 20013 that follow the approach proposed in Corripio (2002) equations 4.23-4.25. The sunrise occurs at $t = 12 \cdot \left(1 - \dfrac{\omega}{\pi}\right)$ and the sunset will be at $t = 12 \cdot \left(1 + \dfrac{\omega}{\pi}\right)$ where $\omega$ is the hour angle. Those equations are based on the assumption that sunrise and sunset occur at the time when the z coordinate of the sun vector equals zero".

Q2) Section 4: There is hardly any discussion of the results. I suggest adding the discussion of the findings in the Result section.

A2) We thank the reviewer for the suggestion. We extended the discussion part as suggested also by reviewer n.2. We added some sentence in the conclusions and some more comments to the results presentation:

"Moreover the Brunt model is able to provide higher performances with the regression model parameters independently of the latitude and longitude classes. For the Idso model the formulation with regression parameter provided lower performances respect to the literature formulation for latitude between [25-30]".

"Although many studies investigated the influence of snow covered area on longwave energy balance (e.g. Plüss and Ohmura, 1997; Sicart et al., 2006), the SMs do not explicitly take into account of it. As presented in König-Langlo Augstein (1994), the effect of snow could be implicitly taken into account by tuning the emissivity parameter"

"Finally, the methodology proposed in this paper provides the basis for further developments such as the possibility to: i) investigate the effect different all-sky emissivity formulation, ii) verify the usefulness of the regression models for locations outside the USA; iii) analyze in a systematic way the uncertainty due to the quality of meteorological input data on the longwave radiation balance in scarce instrumented areas."

Q3) *Section 4.2: I miss a figure or table, which shows the variability of the site-specific model parameters for the different stations analyzed. This information is necessary in order to judge the sensitivity of the parameters on the different climates. Possibly this is reported in the mentioned supplementary material, which I could not find!*

A3) We thank the reviewer for the comment. We attached the missing file of the table containing the parameters value for each model and station few hours after we read the revision. We agree with the reviewer comment and we added below two figures showing the parameters variability for each model and for classes of latitude and longitude.

Figure 1 shows the ratios between the optimal parameter set and the literature parameter set for each model grouped by latitude classes. In general the parameter ratios vary between 0.3 and 2.0 for most of the model and they do not show great variation across latitude classes except model 1, 8, and 9. The same comments are valid for Figure 2 that shows the ratios between the optimal parameter set and the literature parameter set for each model grouped by longitude classes.

For models 1,8, and 9 the ratios reach the maximum value of 6 and for model 1 and 9 they are lower for the latitude classes [25;30] and [30;35] and higher for latitude classes [35;40] and [>40]."

[Figure]

[Figure]

Figure 1: Ratios between optimal and literature parameter set for each model grouped by latitude classes

[Figure]

[Figure]

Figure 2: Ratios between optimal and literature parameter set for each model grouped by longitude classes

Q4) *Section 4.3: You write "you start with optimal parameter set". Is done for every station? Moreover, it might be worth mentioning that the all three parameters of model 10 seem to be quite robust.*

A4) Yes, we started with the optimal parameter set for each station analysed and for each model. We added the following sentence to clarify better:

Old sentence: "The procedure was repeated for each parameter of each model"

New sentence: "The procedure was repeated for each parameter of each model and for each station of the analyzed dataset."

Q5) *Section 4.4: This section is really innovative and therefore its potential needs to be explored more. In practice you often don't have stations nearby, which can be used as a training set. I would like to see how a Ameriflux station in northern Alaska (Arctic) and South America (Tropics) performs with your currently used training set. Is there a specific reason you don't show the*

*RMSE for this section? Which models perform best in this section?*

A5) We thank the reviewer for the comment but we did not considered the two station he-her is referring to. The station in Alaska was excluded because has many no-values in the time-series of downwelling solar radiation compared to the 24 station we considered. The station in Brazil was not considered because we focused our attention in the North America.

Q6) *As I understand the red bars in Figure 8 represent the same KGE values as the bars in Figure 4. A visual test with model 1 shows a disagreement for latitude class 30;35 and 35;40! Please explain.*

A7) We agree with the reviewer comment and we checked again the script to produce Figure 4. We revised the figure and now it is coherent with Figure 8. Here you can find the new figure 4:

[Figure]

Q6) *Section 4.5: Did take into account the soil was snow covered for some time at some stations. Please discuss the effect of snow an your approach and how it influences your results?*

A6) We thank the reviewer for the question. The model parameterizations do not explicitly take into account of the presence of snow on the soil. We agree with the reviewer suggestion to clarify this aspect and we added the following sentence to state it when we present the models:

"Although many studies investigated the influence of snow covered area on longwave energy balance (e.g. Plüss and Ohmura, 1997; Sicart et al., 2006), the SMs do not explicitly take into account of it. As presented in König-Langlo Augstein (1994), the effect of snow could be implicitly taken into account by tuning the emissivity parameter."

Q7) *Section 5: In the Conclusion section, I miss a focus on the actual results, i.e. the evaluation of the different site-specific parameterizations methods and the performance of the different models. For example, it is not enough to write "A broad assessment of the classic longwave radiation parameterizations clearly shows that the Idso (1981) and Brunt (1932) models are the more robust and reliable for all the test sites, confirming previous results". First, I don't "see" this. Please add information based on RMSE or KGE (however this should not be done in the Conclusion section). Second, add the references, which seem to confirm your results.*

A7) We added some comments on the results provided by the Idso and Brunt models, moreover we added the citation of the paper in which this results is confirmed and finally we also commented their performances with the model parameters estimated by the regression models. The new sentence is:

**Minor Comments:**

Q8): L1: Performance of site specific parameterizations:
A8) We revised according the reviewer suggestion. The new title is:
"Performances of site specific parameterizations of longwave radiation"

Q9) L15: for L in SMs
A9) We revised according the reviewer suggestion. The new sentence is:
"to determine by automatic calibration the site-specific parameter sets for L in SMs"

Q10) L29: I guess data also!

A10) We thank the reviewer for the question, but we are not allowed to share data. We provided the website where the ameriflux data are available to download.

Q11) L44: water vapor deficit

A11) We thank the reviewer for the suggestion and we revised the sentence accordingly. The new sentence is:

"To overcome this issue, simplified models (SM), which are based on empirical or physical conceptualizations, have been developed to relate longwave radiation to atmospheric proxy data such as air temperature, water vapor deficit, and shortwave radiation"

Q12) L46: Be consistent - when using L you don't need to add downwelling or upwelling radiation.

A12) We thank the reviewer for the suggestion. In this row we difined hour notation and we indicate the downwelling longwave radiation with the symbol $L_\downarrow$ and the upwelling longwave radiation with the symbol $L_\uparrow$. We used this notation consistently in the whole text.

Q13) L49: Instead of old references I suggest to replace it with newer ones, like doi:10.1007/s00704-012-0675-1 and doi:10.1016/j.coldregions.2013.12.004

A13) We thank the reviewer for the suggestions. We added the newest references as he/she suggested and we preferred to keep the old reference as well.

Q14) L53-54: Why show the results only for this study?

A14) We thank the reviewer for the comment. We show the results of this study because our results partially confirm them.

Q15) L77: Delete "near surface" or replace with "screen level".

A15) We thank the reviewer for the suggestion and we revised accordingly,

deleting "near surface".

Q16) Table 1: The Monteith and Unsworth (1990) is missing in the Reference section, but I guess you mean Unsworth and Monteith (1975) anyway.

A16) We thanks the reviewer for the suggestion and we added the missing citation:

"John Lennox Monteith and MH Unsworth. Principles of Environmental Physics . Butterworth-Heinemann,1990."

Q17) L103-105: Please reformulate. I suggest to make two sentences.

A17) We thank the reviewer for the suggestion. We splitted the sentence in two and the revised sentence is:

"Is well known that surface soil temperature measurements are only available at a few measurement sites. Under the hypothesis that difference between soil and air temperatures is not too big, it is possible to simulate L↑ using the air temperature (Park et al., 2008). "

Q18) Figure 1: "incoming Radiation" in the LWRB box is confusing. Please replace with "Incoming Shortwave Radation".

A18) We thank the reviewer for the suggestion and we revised the figure accordingly. The new figure is presented below:

[Figure]

Q19) L134: Why 0.6. Did you also test other thresholds?

A19) We thank the reviewer for the comment. We tested other thresholds and the one we selected offered a good compromise in effectively detecting clear sky day and in obtaining a time series long enough to be used for calibration purpose.

Q20) L164: Could you please add some information about the used longwave instruments its measurement uncertainties.

A20) The longwave radiation is measured with Eppley Pyrgeometer and the uncertainty is ± 3 W/m2 on average. This information is valid for many stations but some of them changed instrument during the time.

Q21) L182-183: The reason is that the Konzelmann model was calibrated for the Greenland ice sheet, which has a totally different climate than you stations.

A21) We thank the reviewer for the comment and we modified the sencente according his/her suggestion:

New sentence: "Model 8 (Konzelmann et al. (1994)) does not perform very well for many of the stations likely because the model parameters were estimated for the Greenland where the ice plays a fundamental role on the energy balance."

Q22) L225: For better understanding please link this part to the former section by changing the first sentence to: The just performed calibration procedure to estimate: : :
requires: : :
A22) We thank the reviewer for the comment and we modified the sentence according his/her suggestion:
New senetence: "The just performed calibration procedure to estimate the site specific parameters for L $\downarrow$ models requires measured downwelling longwave data."

Q23) L232: The URL is invalid: I suggest to add this information also to the supplementary material.
A23) We thank the reviewer for the suggestion and we are going to update the link and submit the regression R script in the supplementary material.

Q25) L244: figures (8) and (9)
A25) We thank the reviewer and we revised the typo according his suggestion.

References

König-Langlo, G., & Augstein, E. (1994). Parameterization of the downward long-wave radiation at the Earth's surface in polar regions. *Meteorologische zeitschrift, NF 3, Jg. 1994, H. 6*, 343-347.

Sicart, J. E., Pomeroy, J. W., Essery, R. L. H., & Bewley, D. (2006). Incoming longwave radiation to melting snow: observations, sensitivity

and estimation in northern environments. *Hydrological processes*, *20*(17), 3697-3708.

Plüss, C., & Ohmura, A. (1997). Longwave radiation on snow-covered mountainous surfaces. *Journal of Applied Meteorology*, *36*(6), 818-824.

---

## Author Comment (AC3) · 12 Aug 2016

**General comments:**

*The study analyses the performance of 10 empirical parameterizations of incoming longwave radiation with original parameters, site-specific fitted parameters and parameters obtained from regression with average climate variables. The calibration and validation data is taken from the AmeriFlux network. Additionally, the study compares the accuracy of outgoing longwave radiation estimates using soil temperature, soil surface temperature and air temperature. In most parts, the study repeats a similar analysis as other papers (Flerchinger et al., 2009; Juszak and Pellicciotti, 2013; Carmona et al., 2014), which is the comparison of parameterizations of incoming longwave radiation with original and fitted parameters.*

*The novelty of the study arises from the site-specific estimation of the parameters using multivariate linear regression. This part is interesting for future studies that do not have longwave radiation data available. As the multivariate linear regression is the new and relevant part of the study, Section 4.4 should be elaborated more and presented in more detail. If this part is emphasized strongly, the paper may be published in HESS after major revisions.*

The authors thank the reviewer for the useful suggestions and corrections he provided in the revision. Below, we answered point by point to each of them.

**Q1)** *The results section mixes methods, results and discussion. The methods should be moved to the methods section, the discussion should be separate and longer to incorporate (i) Which models are best at all sites and when used with parameter estimates from the regression approach? (ii) Are the regressions likely to work outside the USA? (iii) What are possible sources of uncertainty?*

**A1)** We thank the reviewer for the suggestion. We modified the structure of the paper according his suggestions. In the revised paper, besides the subsections Calibration (2.1) and Verification (2.2), we added two subsections that describe the model sensitivity analysis (2.3) and the multi-regression model method (2.4). These two subsections previously were located in the results section. We prefer to comment and discuss the results in the subsection where we presented them. This allows us to not completely modify the original structure of the paper. Moreover we added new sentences in the conclusion section containing information in line with the reviewer suggestions. The new sentences are:

"Moreover the Brunt model is able to provide higher performances with the regression model parameters independently of the latitude and longitude classes. For the Idso model the formulation with regression parameter provided lower performances respect to the literature formulation for latitude between [25-30]".

"Although many studies investigated the influence of snow covered area on longwave energy balance (e.g. Plüss and Ohmura, 1997; Sicart et al., 2006), the SMs do not explicitly take into account of it. As presented in König-Langlo Augstein (1994), the effect of snow could be implicitly taken into account by tuning the emissivity parameter"

"Finally, the methodology proposed in this paper provides the basis for further developments such as the possibility to: i) investigate the effect different all-sky emissivity formulation, ii) verify the usefulness of the regression models for locations outside the USA; iii) analyze in a systematic way the uncertainty due to the quality of meteorological input data on the longwave radiation balance in scarce instrumented areas."

**Q2).** *Most formulas have either not been cited correctly (Table 1 of the manuscript), or the given empirical parameters (Table 2 of the manuscript) were derived for different units of the input variables and can thus not be used with other units and without adjustment. This is a serious issue as it affects the results and conclusions. It should be corrected and all graphs need to be updated. Also some of the conclusions like "Model 8 (Konzelmann et al. (1994)) does not perform very well for some reason." (Line 182) and "Regarding the L # simulations, the Brunt (1932) and Idso (1981) SMs, in their literature formulations, provide the best performances in many of the sites." (Abstract) may be wrong.*

**A2)** We thank the reviewer for the precious suggestion. We double-checked each formula and each unit both in the paper and in the source code. What we found was that only one model was implemented with one imperfection (Model 8). We revised it and we re-executed all the simulations: literature parameters, calibrations, sensitivity, and we re compute the regressions model. The other models were correctly implemented but imperfectly presented in the table 1. We revised the table as well. In the specific comments we answered point by point to each of the reviewer's comment about the models.

**Q3)** *Some of the cited literature does not appear in the references.*
**A3)** We double-checked again the cited literature.

**Q4)** *c is used for the clearness index and the cloud cover fraction. Please rename one of them and write the equation to convert them.*
**A4)** We clarified the difference between the cloud cover fraction (c) and the clearness index (s) in the revised version of the paper. The modified sentences specify those differences and how the two indices are related each other:
Sentence 1: "where c [-] is the cloud cover fraction and a [-] and b [-] are two calibration coefficients."
Sentence 2: "In this study we use the formulation presented in Campbell (1985) and Flerchinger (2000), where c is related to the clearness index s [-], i.e. the ratio between the measured incoming solar radiation, $I_m$ [Wm$^{-2}$], and

the theoretical solar radiation computed at the top of the atmosphere, $I_{top}$ [$Wm^{-2}$], according the following relationship: c=1-s, (Crawford and Duchon, 1999)."

**Q5)** *State in more detail the results of the parameter estimate by regression and provide the formula for the best model including average parameters*.

Q6) We thank the reviewer for the suggestion. We were thinking that, providing the reader or the user, is mostly interested on having a tool that receive in input the average annual rainfall, air temperature, relative humidity and the elevation of the site, and get a parameter set for a selected model. This is the reason why we provided the link to the R-cran source code to perform this operation. Presenting in a table the value of the regression parameters for each model was not our focus but we added it in a supplementary material.

**Specific comments:**

**Q6)** *Abstract The study described in the manuscript is largely independent of the hydrological model JGrass-NewAge. The authors do not present any results concerning hydrology. Thus, this model should not be central in the first sentences of the abstract.*

**A6)** We agree with the reviewer comment on the fact that we do not present anything about hydrology and hydrological simulation with NewAge. Actually the components we implemented in this paper are compatible with the existing NewAge components (such as shortwave radiation model, snow model) and can be connected each other. Moreover in this paper we used some of NewAge capability such as: i) the input output managing (reading the shapefiles, the digital elevation model, the time series), ii) the shortwave radiation model,and iii) the optimization algorithm. For this reason we would like to preserve JGrass-NewAge in the abstract but, in order to satisfy the reviewer suggestion, we removed the sentence "hydrological model JGrass-NewAge" and we used the sentence "JGrass-NewAge modeling system". The new sentence is:

"In this work ten algorithms for estimating downwelling longwave atmospheric

radiation (L$\downarrow$) and one for upwelling longwave radiation (L$\uparrow$) are integrated into the JGrass-NewAge modeling system."

**Q7)** *L13–15 These are really 3 points: (i) original formulation, (ii) site specific calibration, and (iii) parameter estimation based on average site characteristics.*
**A7)** We agree with the reviewer comments on the fact that they are three points but, the third point "*parameter estimation based on average site characteristics*" is explained after the period: "For locations where calibration is not possible because of a lack of measured data, we perform a multiple regression using on-site variables, such as mean annual air temperature, relative humidity, precipitation, and altitude". This allows us to better explain the importance of the regression because in most of the case calibration is not possible.

**Q8)** *L16 Name all variables instead of 'such as'*
**A8)** We actually named all the variables, for this reason we removed the word "such as" and we used "i.e.". The new sentence is:
"For locations where calibration is not possible because of a lack of measured data, we perform a multiple regression using on-site variables, i.e. mean annual air temperature, relative humidity, precipitation, and altitude"

**Q9)**. L21–23 This conclusion may change with correct model formulation. The relative performance of the models should be discussed in more detail in a discussion section.
**A9)** The conclusions did not dramatically changed after the models' revision. We slightly modified them according the new results. In particular the main change regards only the model 8 in the sense that after the modification the performances using the literature formulation improved respect the results presented in the previous version of the paper. On the other side the optimal parameter values did not change dramatically and remain of the same order of magnitude of the previous version of the paper.

**Q10)** *L29 Remove this sentence.*

**A10)** We removed the following sentence according the reviewer suggestion: "Models and regression parameters are available for any use, as specified in the paper."

**Q11).** *L31 3-100 (1 is still shortwave radiation (Oke, 1987))*

A11) We modified the sentence according the reviewer suggestion. The new sentence is:

"Longwave radiation (4-100 μm) is an important component of the radiation balance on earth and it affects many phenomena"

**Q12)** *L34 Remove 'very expensive', that is relative*

**A12)** We removed "very expensive" as suggested by the reviewer. The new sentence is:

"Longwave radiation is usually measured with pyrgeometers, but these are not normally available in basic meteorological stations,"

**Q13)** *L58–59 I do not agree with this major advantage of the current study as compared to the former studies. The empirical formulations of longwave radiation are very simple equations that can be included easily in any model without the need of an open-source tool. Instead, the authors could refer to their parameter estimation approach: 'However, none of the above studies have developed a method to estimate the parameters for any location based on basic site characteristics and ready for practical use by other researchers and practitioners.' More sentences on the added value of this study are needed. What are the research questions?*

**A13)** We thank the reviewer for the suggestion and we modified the sentence underling the importance of the study in terms of providing a systematic estimate of site-specific model parameters in the location where it is possible and their estimate with a regression model where this is not possible. Finally we really would like to preserve the importance of providing open-source tools ready to be downloaded and eventually used for a reproducible research. The modified sentence we added in the revised paper is:

"However, none of the above studies have: i) developed a method to

systematically compute the site-specific model parameters for location where measurements are available, and ii) provided their estimate for any location based on basic site characteristics. Moreover, differently from other studies, all the tools used in this paper are pen-source, well documented, and ready for practical use by other researchers and practitioners."

**Q14)** *L68–74 Paragraph not needed*
**A14)** We deleted the paragraph as suggested by the reviewer

**Q15)** *L77 the 'k' of 'kg' should be lower-case; it would be more intuitive to provide the unit Wm$^{-2}$ K$^{-4}$ as L is given in Wm$^{-2}$*
**A15)** We agree with the reviewer comment and we modified the units of the Boltzman constant as he suggested. The new sentence is:
"where σ = 5.670·10−8 [W m$^{-2}$ K$^{-4}$]"

**Q16)** *eq. 3 It should be noted that this equation was proposed by Bolz (1949), and that there are other options that potentially work better (Flerchinger et al., 2009; Juszak and Pellicciotti, 2013). The authors should consider using Unsworth and Monteith (1975), which was recommended in both studies.*
A16) We thank the reviewer for the suggestion and we added some more sentence on the possibility to use other formulation respect to equation 3. Testing different formulation of equation 3 was not the object of this study and the flexibility of the system allow the user to add a new formulation for cloudy sky conditions and preserve all the other part of the code we shared as open-source. Unfortunately we preferred to keep equation 3 and we cited a couple of papers where it was used. Moreover, as the reviewer suggested we added the following sentence to clarify the possibility to use other formulations and that those could work better in some cases. The new sentence is:
"The formulation presented in equation 3 was proposed by Bolz (1949) applied in other studies (Carmona et al., 2013, Maykut and Church, 1973, Jacobs, 1978). Evaluating the effectiveness of different formulations respect to equation 3 is still an open question which is not object of the current paper. It has been investigated in several studies (i.e. Flerchinger et al., 2009,

Juszak and Pellicciotti, 2013) and some of them recommended the one proposed by Unsworth and Monteith (1975). "

**Q17)** *L81 c is not the clearness index but the cloud cover fraction (as in line 84)*

**A17)** We revised the sentence according the reviewer suggestions. The new revised sentences are:

Sentence 1: "where c [-] is the cloud cover fraction"

Sentence 2: "The cloud cover fraction, c, can be estimated from solar radiation measurements"

**Q18)** *L87 Related how? Provide equation!*

**A18)** We modified the sentence providing the equation as requested by the reviewer. The modified sentence is:

"In this study we use the formulation presented in Campbell (1985) and Flerchinger (2000), where c is related to the clearness index s [-], i.e. the ratio between the measured incoming solar radiation, $I_m$ [$Wm^{-2}$], and the theoretical solar radiation computed at the top of the atmosphere, $I_{top}$ [$Wm^{-2}$], according the following relationship: c=1-s, (Crawford and Duchon, 1999)"

**Q19)** Table 1 I have doubts that all formulas in Table 1 are correct and that the parameters in Table 2 have been adjusted to the units of water vapour pressure (and in some cases radiation). I suggest you check Juszak and Pellicciotti (2013) for adjusted parameters. More specifically please consider:

1. Angstrom [1918] does not appear in the reference list. Please provide the correct reference and check the original publication or cite the paper you took the parameters from. Did you adjust the original parameters to match the units in which you computed the radiation and inserted humidity and temperature? I have doubts in the Angstrom case where one original publication computes the radiation in cal $cm^{-2}$ $min^{-1}$ (Ångström, 1916). Ångström (1916) also uses $e^{Ze}$ instead of $10^{Ze}$.

2. Brunt (1932) uses water vapour pressure in millibar not kPa. Did you adjust the parameter Y?

3. Swinbank (1963) is clearly used wrongly. The parameters provided in

Table 2 do not refer to the clear sky emissivity but to a formula that computes the radiation directly, and in m W cm$^{-2}$.

4. Brutsaert (1975) uses water vapour pressure in millibar not kPa. Please adjust the parameters X and Y.

5. Monteith and Unsworth [1990] does not appear in the literature list. Please double-check the formula and parameters and provide the correct citation.

6. Konzelmann et al. (1994) uses water vapour pressure in Pa not kPa. Please adjust the parameters X and Y.

7. Dilley and O'Brien (1998) uses the given formula (Table 1) with the parameters (Table 2) to directly compute the longwave flux, not the emissivity. To get the emissivity, the formula has to be divided by $\sigma$ T$^4$ Use round brackets for the reference year as in the rest of the manuscript.

A19) We thank the reviewer for the suggestion. We referred all our formula to the paper Flerchinger et al., 2009 and in particular we follow the table 1 of the paper. In our Table 1 we forgot the specify some of the footnotes presented in the Table 1 of the Flerchinger et al., 2009 paper. For this reason many of the reviewer comments were just related to the fact that the Table 1 in our paper was not completely correct but the code it is. Thanks to the author comments we double-checked all the formulations and we realized that one model was implemented wrong in the sense that in Konzelmann the vapour pressure was in kPa and not in Pa as it should be. We modified the model and re executed the simulation for all the stations (literature formulation, calibration, sensitivity, and regression). We moreover answered point by point to the reviewer comments below:

1) We implemented the model in the right way as specified in the Flerchinger et al., 2009 paper but we forgot to specify that the version was the implemented in Niemela et al. [2001]. We added it as footnotes in the Table 1.

2) We implemented the model in the right way as specified in the Flerchinger et al., 2009 paper but we forgot to specify that the version was the implemented in Niemela et al. [2001]. We added it as footnotes

in the Table 1.

3) We implemented the model as specified in Flerchinger et al., 2009, but we forgot to divide by $\sigma T^4$ to obtain the emissivity. We modified the table accordingly.

4) We implemented the model as specified in Flerchinger et al., 2009 and we cite it.

5) We double-checked the formula and we added the following correct citation:

   "Monteith JL, Unsworth MH (1990) Principles of Environmental Physics Edward Arnold, London"

6) We revised the model according the reviewer suggestion and we repeated all the computation for this model. Fortunately there was not a dramatic change in the model results and model parameters. We modified the discussion of the results according the new model output.

7) We implemented the model as specified in Flerchinger et al., 2009, but we forgot to divide by $\sigma T^4$ to obtain the emissivity. We modified the table accordingly.

**Q20)** *L116 No one-sentence paragraph, this sentence can be removed.*

**A20)** We removed the sentence as the reviewer suggested.

**Q21)** *Figure 1 How do Im and Itop fit into this schematic? Only those variables are explained later in the text. The 'Modelled longwave radiation' and 'Measured longwave radiation' items in the Verification box are wrongly connected. Is the SWBR always modelled? Does that affect the optimisation process?*

**A21)** We revised the figure according the reviewer suggestions i.e. correcting the connections between measured and modeled radiation for the verification box and specifying what is $I_{top}$ and $I_m$ explicitly and the figure and not only in the text. Below you can find the revised figure.

[Figure]

**Q22)** *L134 Did you try different thresholds? 0.6 seems quite low. Did you verify that you do not include cloudy or partly cloudy observations in the clear sky calibration? If you calibrate $e_{clear}$ at c = 0.6, $e_{all\_sky}$ at that condition will be wrong as you compute it from $e_{all\_sky} = e_{clear} (1 + a\ c^b)$ and c=0.6.*

**A22)** The choice of the threshold was due to two main reasons: firstly we considered some values from literature to define clear sky conditions and we find that they vary between 0.6 and 0.7 (Li et al., 2001; Okogbue et al., 2009); secondly we tried different threshold and 0.6 provided a good compromise to get equally long time series of measured downwelling clear radiation for all the stations. This was important for a reliable calibration process. Of course the reviewer comments on the emissivity in all sky condition is correct and we specified it in the revised paper adding the following sentence:

"On one side, a threshold of 0.6 to define the clear-sky conditions helps in the sense that allow to define time-series of measured clear-sky L ↓ with comparable length in all the stations, and this is useful for a reliable calibration process. On the other side, it introduces an error in computing the emissivity

in all-sky condition using equation 3 which could be compensated by the optimization of the parameters a and b."

**Q23)** L143–144 Please provide all variables (not 'such as'); altitude is not a climatic variable.

**A23)** We thank the reviewer for the suggestion and we revised the sentence accordingly. The new sentence is:

"As well as parameter calibration, we carry out a model parameter sensitivity analysis and we provide a linear regression model relating a set of site-specific optimal parameters with mean air temperature, relative humidity, precipitation, and altitude."

**Q24)** *L156 What is N?*

**A24)** N is the length of the measured and modeled time-series. We added the following sentence to the revised paper:

"where M and S represents the measured and simulated time-series respectively and N is their length."

**Q25)** *L161–162 Sentence not relevant, remove it.*

**A25)** We thank the reviewer for the suggestion. We agree in part with him: we modified the sentence as he ask, but we would like to keep it in order to give visibility to other works that used the same dataset.

Old sentence: "The dataset is widely known and used for biological and environmental applications. To cite a few, Xiao et al. (2010) used Ameriflux data in a study on gross primary production data, Kelliher et al. (2004) in a study on carbon mineralization, and Barr et al. (2012) in a study on hurricanes."

New sentence: "The dataset is well-known and used in several applications such as Xiao et al. (2010), Kelliher et al. (2004), and Barr et al. (2012)."

**Q26)** *L166–168 There is also a gradient towards the colder climate. Why did you choose these 24 stations and not all stations?*

**A26)** Among the stations where the model input data were available we selected the 24 as a good compromise between the goals of: i) covering

different climates, ii) ensuring a reasonable quality of the data (avoiding long no-value periods in the time-series), and iii) ensuring a reliable computational time for models calibration and verification.

**Q27)** *Figure 2 Use same index for stations as in Table 4 and make the index bigger so it is readable.*

**A27)** We thank the reviewer for the suggestion and we modified the figure according his-her suggestion. The new figure is:

[Figure]

**Q28)** *Table 4 How was the climate defined? 'mild' and 'strongly seasonal' do not match the classic categories.*

**A28)** The classification is defined for each station of the Ameriflux network and it is a standard classification. More information on the classification are available at the web page of each station (https://fluxnet.ornl.gov/site/833 for the station 833)

**Q29)** *Section 4.1 Update section with correct model implementation and parameters.*

**A29)** As above specified, we re-executed the simulations for the model 8 we re-plot the results. All figures have been updated.

**Q30)** *Figure 3 Name models in the caption*

**A30)** We agree with the reviewer suggestion and we modified the figure accordingly. The new figure is reported below:

[Figure]

**Q31)** *Figures 4–6, 8–9 Use boxplots instead of barplots to show the variability within the groups and the range of variation. Reorganise the content to have only two Figures: one for clear sky and one for all sky. In both figures, boxes for results of (i) original parameters, (ii) fitted parameters, and (iii) parameters from regression analysis should be next to each other to enable direct comparison. The figures can be arranged in subplots either one per model, or one per latitude / longitude class. Please choose colours that allow black+white printing and consider color-blind people.*

**A31)** We thank the reviewer for comment. We agree in part with him: we prefer to keep the plots as we made because it was an original idea of all the coauthors and the meaning of not reducing everything to 2 figures was because we wanted to facilitate the reader and his comprehension of the results. We believe that this configuration was a good compromise between the amount of information for each plot and the possibility of a common reader to easily get the results. We strongly agree with the reviewer to modify the scale color and facilitate color-blind people. We used the r cran package ggthemes and the function scale_fill_colorblind() to re plot the results of the figures in discussion. Below you can find the final figures:

[Figure]

[Figure]

[Figure]

[Figure]

**Q32)** *Section 4.2 Update section with correct model implementation and original parameters.*

**A32)** As we specified before, all the plots have been updated.

**Q33)** *L201–202 This should be moved and discussed in more detail in a discussion section.*

**A33)** We thank the reviewer for the comment. As we explained in the answer A1) we prefer to keep the discussion of the results in the same section in which the results are presented. This is further justified in the answer A1.

**Q34)** *L213 Time series from which station? Was the analysis done for all stations?*

**A34)** We thank the reviewer for the suggestion. Yes the procedure was repeated for each station and we specified it in the revised paper:

Old sentence: "The procedure was repeated for each parameter of each model"

New sentence: "The procedure was repeated for each parameter of each model and for each station."

**Q35)** *L206–214 This belongs to the methods section.*

**A35)** We agree with the reviewer suggestion and, as specified in the answer

A1) we moved that part in the methods section as subsection.

**Q36)** *Figure 7 Given the methods description, why is the peak not always in the middle of the parameter range? Caption: 'of' is missing an 'f'; describe the meaning of the boxes and the line!*

**A36)** We thank the reviewer for the comment. In general the peak is corresponding to the optimal parameter set. Small variations are possibly due to the fact that we subdivided the range of a given parameter into ten equal-sized classes and for each class the corresponding KGE values are presented as a boxplot. This approximation can influence the shape of the final plot. Moreover we agree with the reviewer comment on the figure and we revised the caption accordingly. The new caption is:

"Results o the model parameters sensitivity analysis. It presents as boxplot the variation of the model performances due to a variation of one of the optimal parameter and assuming constant the others. The procedure is repeated for each model and the blue line represents the smooth line passing through the boxplot medians."

**Q37)** L225–243 This belongs to the methods section.

**A37)** We agree with the reviewer suggestion and, as specified in the answer A1) we moved that part in the methods section as subsection.

**Q38)** *Equation 8 Do not use 'a' as it is used for something else in Equation 4*

**A38)** We agree with the reviewer suggestion and we used i, that stand for intercept, instead of a.

**Q39)** *L244–250 Compare also with fitted parameters.*

**A39)** We thank the reviewer for the comment. We were thinking to plot also the optimal parameters results but at the end we decided to keep only the regression and the literature results, for two reasons: the first is the the innovation of this section is to provide a method (the regression) that does better than (or at least equal to) the literature formulation, so regression and literature are fundamental in the plot, the second is that the results with optimal parameter set have been presented in the previous section.

**Q40)** *Section 4.4 Update section with correct model implementation and original parameters.*

**A40)** As we specified before, all the plots have been updated.

**Q41)** *L267–269 This should be moved and discussed in more detail in a discussion section. How about snow cover? How about the different latitudes?*

**A41)** We thank the reviewer for the comment. As we explained in the answer A1) we prefer to keep the discussion of the results in the same section in which the results are presented. This is further justified in the answer A1. Regarding the snow cover we added the following sentence, as requested also by the reviewer n.1:

"Although many studies investigated the influence of snow covered area on longwave energy balance (e.g. Plüss and Ohmura, 1997; Sicart et al., 2006), the SMs do not explicitly take into account of it. As presented in König-Langlo Augstein (1994), the effect of snow could be implicitly taken into account by tuning the emissivity parameter."

**Q42)** *Conclusions Update section with correct model implementation and original parameters.*

A42) We thank the reviewer for the suggestion and we revised the conclusion section accordingly.

**Q43)** *Supplementary material Please use the same station IDs as in the manuscript. Please include the detailed results of the parameter regression.*

**A43)** We thank the reviewer for the suggestion and we revised the supplementary material section accordingly. Moreover in the supplementary material we provided the link to the R-cran code to estimate the regression parameters given the input data.

Technical corrections:

**Q44)** *L12 24 instead of twenty-four*

**A44)** We used 24 instead of twenty-four in the whole text of the revised paper

**Q45)** *L36 put references in brackets*

**A45)** We revised according the reviewer suggestion.

**Q46)** *L40 put references in brackets*

**A46)** We revised according the reviewer suggestion.

**Q47)** *L51 'They' instead of 'It'*

**A47)** We revised according the reviewer suggestion.

**Q48)** *L52 remove 'so'*

A48) We revised according the reviewer suggestion.

**Q49)** *L64 put reference in brackets*

**A49)** We revised according the reviewer suggestion.

**Q50)** *Table 1 caption: units not in italics*

**A50)** We revised according the reviewer suggestion.

**Q51)**L101 space missing before reference

**A51)** We revised according the reviewer suggestion.

**Q52)** *L104 put references in brackets*

**A52)** We revised according the reviewer suggestion.

**Q53)** *L106 replace ';' with ','*

**A53)** We revised according the reviewer suggestion.

**Q54)** *L122 remove brackets from reference*

**A54)** We revised according the reviewer suggestion.

**Q55)** *L158 24 instead of twenty-four*

**A55)** We revised according the reviewer suggestion.

**Q56)** *L165 24 instead of twenty-four*

**A56)** We revised according the reviewer suggestion.

**Q57)** *L178 1:1 instead of 45 degree*

**A57)** We revised according the reviewer suggestion.

**Q58)** *L219 'around the' instead of 'about'*

**A58)** We revised according the reviewer suggestion.

**Q59)** *L231 'supplementary' instead of 'complementary'*

**A59)** We revised according the reviewer suggestion.

**Q60)** *L244 Figure 10 shows something else*

**A60)** We revised according the reviewer suggestion.

**Q61)** *L275 24 instead of twenty-four*

**A61)** We revised according the reviewer suggestion.

**Q62)** *L284 24 instead of twenty-four*

**A62)** We revised according the reviewer suggestion.

**Q63)** *L303 Reformulate 'In order that'*

**A63)** We modified the sentence as suggested by the reviewer. The new sentence is:

"Researchers interested in replicating or extending our results are invited to download our codes at"

References

Flerchinger, G. N., Xaio, W., Marks, D., Sauer, T. J., & Yu, Q. (2009). Comparison of algorithms for incoming atmospheric long‑wave radiation. *Water Resources Research*, *45*(3).

Li, Danny HW, and Joseph C. Lam. "An analysis of climatic parameters and sky condition classification." *Building and Environment* 36, no. 4 (2001): 435-445.

Okogbue, E. C., J. A. Adedokun, and B. Holmgren. "Hourly and daily clearness index and diffuse fraction at a tropical station, Ile‑Ife, Nigeria." *International Journal of Climatology* 29, no. 8 (2009): 1035-1047.

---

## Referee Report (RR1)

[referee-annotated manuscript omitted]

---

## Author Response (AR2)

1Q) (4-100) remove, because the spectrum goes beyond 4 and is intersecting with the shortwave.

1A) We thank the reviewer for the suggestion and we fixed it.

2Q) across the USA: contiguous2A) We thank the reviewer for the suggestion and we fixed it.across the contiguous USA

3Q) the contiguous USA: contiguous3A) We thank the reviewer for the suggestion and we fixed it.

4Q) Please add the following sentence: Longwave radiation was measured with Eppley Pyrgeometers with uncertainty of +/- 3 W/m-2.4A) We thank the reviewer for the suggestion and we added it.

5Q) "where the ice plays a fundamental role": Modify ice with snow and ice5A) We thank the reviewer for the suggestion and we modified it:"where snow and ice play a fundamental role"

6Q) Instead of Moreover specify: "Regarding longwave downwelling radiation the"

6A) We thank the reviewer for the suggestion and we modified the sencence accordinly.

7Q) "model is able to provide higher performances": higher than what?7A)We thank the reviewer for the suggestion and we revised the sentence:

"model is able to provide on average the best performances with the regression model parameters independently of the latitude and longitude classes"

8Q) lower performances respect to the: use with respect to

8A) We thank the reviewer for the suggestion and we revised the sentence.

9Q) the effect different all-sky emissivity: the effect of

9A) We thank the reviewer for the suggestion and we revised the sentence.

10Q) the regression models for locations outside: use climates

10A) We thank the reviewer for the suggestion and we revised the sentence.

11Q) outside the USA: use contiguous

11A) We thank the reviewer for the suggestion and we revised the sentence.

**The organization of the paper has improved and the methods section is more comprehensive now.**

We thank the reviewer for the comments that have improved again the quality of our paper. Below you can find the point-by-point answers to the questionssuggestions:

However, otherwise many of the issues in the first manuscript version still persist in the revised version, for example:

**Q1) Unlike the authors answer (2nd reviewer, A3), Angstrom (1918) still does not appear in the reference list.**

A1) We verified the reference and we have corrected it in the revised paper.

**Q2) The mix of citation styles in both, the text and the reference list. The reference list is not sorted alphabetically.**

A2) We verified and changed the style all the reference in the new version of the paper.

Q3) Unlike the authors answer (2nd reviewer, A27), Figure 2 still does not show the locations of the tests sites with the correct index. The numbers should be between 1 and 24, and all numbers should be visible.

A3) We modified the figure in order to make all the number visible. Below you can find the final figure:

---

## Author Response (AR3)

**Editor comments**

**Editor Decision: Publish subject to minor revisions (further review by Editor)** (28 Oct 2016) by Prof. Bettina Schaefli
Comments to the Author:

Dear Authors
the manuscript is now almost ready for publication. Given the numerous "technical mistakes" that were left after the first revision (see reviewer comment 2) and the numerous citation style mistakes that are still left, I strongly recommend you check the manuscript again carefully. In particular, you should check all the citations (each instance) and correct the latex citation style (citep / citet).

**We thank the editor for the suggestions: we re-checked the text and we revised again all the reference in the text. Below you can find the reply to the specific comments:**

Examples of wrong style are:

1Q) line 37: Augustine et al. (2000) ,Augustine et al. (2005) and Baldocchi et al.
(2001) (citet instead of citep)
**1A) We thank the editor and we revised accordingly**

2Q) line 41: (e.g. Key and Schweiger (1998), Kneizys et al. (1988)) (citet instead of citep)
**2A) We thank the editor and we revised accordingly**

3Q) line 66: Formetta et al. (2014a). (citet instead of citep)

**3A) We thank the editor and we revised accordingly**

4Q) line 143: (KGE, Gupta et al. (2009) ) (citet instead of citep)

**4A) We thank the editor and we revised accordingly**